# Quantitative Proteomics and Molecular Mechanisms of Non-Hodgkin Lymphoma Mice Treated with Incomptine A, Part II

**DOI:** 10.3390/ph18020242

**Published:** 2025-02-11

**Authors:** Normand García-Hernández, Fernando Calzada, Elihú Bautista, José Manuel Sánchez-López, Miguel Valdes, Marta Elena Hernández-Caballero, Rosa María Ordoñez-Razo

**Affiliations:** 1Unidad de Investigación Médica en Genética Humana, UMAE Hospital Pediatría 2° Piso, Centro Médico Nacional Siglo XXI, Instituto Mexicano del Seguro Social, Av. Cuauhtémoc 330, Col. Doctores, Mexico City 06725, Mexico; romaorr@yahoo.com.mx; 2Unidad de Investigación Médica en Farmacología, UMAE Hospital de Especialidades, 2° Piso CORSE, Centro Médico Nacional Siglo XXI, Instituto Mexicano del Seguro Social, Av. Cuauhtémoc 330, Col. Doctores, Mexico City 06725, Mexico; valdesguevaramiguel@gmail.com; 3SECIHTI-División de Biología Molecular, Instituto Potosino de Investigación Científica y Tecnológica A.C., San Luis Potosí 78216, San Luis Potosí, Mexico; francisco.bautista@ipicyt.edu.mx; 4Hospital Infantil de Tlaxcala, Investigación y Enseñanza, 20 de Noviembre S/M, San Matias Tepetomatitlan, Apetatitlan de de Antonio Carvajal 90606, Tlaxcala, Mexico; cienciaflosan@gmail.com; 5Phagocytes Architecture and Dynamics, IPBS, UMR5089 CNRS-Université Toulouse 3, 205 route de Narbonne, 31077 Toulouse, France; 6Sección de Estudios de Posgrado e Investigación, Escuela Superior de Medicina, Instituto Politécnico Nacional, Plan de San Luis y Salvador Díaz Mirón S/N, Col. Casco de Santo Tomás, Miguel Hidalgo, Mexico City 11340, Mexico; 7Facultad de Medicina, Biomedicina, Benemérita Universidad Autónoma de Puebla, Puebla 72410, Puebla, Mexico; ehdezc@yahoo.com

**Keywords:** incomptine A, non-Hodgkin lymphoma, proteome TMT-based, molecular docking, necroptosis

## Abstract

**Background/Objectives:** Incomptine A (IA) has cytotoxic activity in non-Hodgkin lymphoma (NHL) cancer cell lines. Its effects on U-937 cells include induction of apoptosis, production of reactive oxygen species, and inhibition of glycolytic enzymes. We examined the altered protein levels present in the lymph nodes of an in vivo mouse model. **Methods:** We induced an in vivo model with Balb/c mice with U-937 cells and treated it with IA or methotrexate, as well as healthy mice. We determined expressed proteins by TMT based on the LC-MS/MS method (Data are available via ProteomeXchange with identifier PXD060392) and a molecular docking study targeting 15 deregulated proteins. We developed analyses through the KEGG, Reactome, and Gene Ontology databases. **Results:** A total of 2717 proteins from the axillary and inguinal lymph nodes were analyzed and compared with healthy mice. Of 412 differentially expressed proteins, 132 were overexpressed (FC ≥ 1.5) and 117 were underexpressed (FC ≤ 0.67). This altered expression was associated with 20 significantly enriched processes, including chromatin remodeling, transcription, translation, metabolic and energetic processes, oxidative phosphorylation, glycolysis/gluconeogenesis, cell proliferation, cytoskeletal organization, and with cell death with necroptosis. **Conclusions:** We confirmed the previously observed dose-dependent effect of IA as a secondary metabolite with important potential as an anticancer agent for the treatment of NHL, showing that the type of drug or the anatomical location influences the response to treatment. The IA promises to be a likely safer and more effective treatment to improve outcomes, reduce toxicities, and improve survival in patients with NHL, initially targeting histones and transcription factors that will affect cell death proteins.

## 1. Introduction

Cancer contributions to early mortality in the world, if incidence trends continue, will be doubling cases by 2070 relative to 2020 [1]. The most common is adult non-Hodgkin lymphoma (NHL), with 553,010 cases and 250,475 deaths around the world in 2022 [2]. In Mexico, NHL is the ninth cause of cancer for both genders, and the fourth cause of mortality among malignant tumors of lymphatic tissue. NHL is a heterogeneous group of lymphoid tissue neoplasms, arising in the lymph nodes or tissues due to malignant transformation of B/T lineage, and NK lymphocytes, characterized by an increased proliferation and/or reduction in apoptosis [3,4,5,6].

Furthermore, cancer prevention and control, as well as access to new treatments, could change the prognostic in Mexican patients. Anticancer drugs used in chemotherapy show several side effects, such as methotrexate (Figure 1), which leads to genotoxic damage, which is the drug of choice for the treatment of NHL in Mexico [6,7,8,9]. Understanding the biology of NHL and searching for ideal treatment that is safer and more effective leads to the target and discovery of specific anti-lymphoma novel agents. Incorporating biology and proteomic studies of NHL into therapeutic studies, leading to improved outcomes or reduced toxicities [10,11].

Incomptine A (IA), a heliangolide-type sesquiterpene lactone (SL) isolated from leaves of *Decachaeta incompta* (Figure 1), with phytotoxic, antiprotozoal, antibacterial properties, similar to other SL, exhibits anticancer effects through down-regulation of Bcl-2 family anti-apoptotic proteins, reduced glutathione depletion, redox cell balance changes, and impacts on NF-kB, STAT3, or signaling pathways. These activities affect protein expression in cell cycle, proliferation, metastasis, or cellular invasion, and the production of free radicals induces apoptosis [12].

Previously, we discovered that IA possesses pharmacological potential as an anticancer agent in U-937 [5], we discerned whether IA in vitro regulates oncogenic pathways, and we found that it had cytotoxic activity in non-Hodgkin lymphoma subtypes, as IA induce apoptosis in U-937 cells due redox imbalance. We also found inhibition of glycolytic enzymes (LDHA, LDHB, and ALDOA) and cytoskeleton proteins that might interact with IA as their targets and promote an antitumor effect or cell death by energy metabolism disruption [13]. Additionally, IA showed a cytotoxic effect dose-dependent in leukemia cell lines (REH, HL-60, and K-562) [14].

To confirm IA potential use as a therapeutic agent and antitumor activity against NHL modifying pathways and/or cellular processes, the aim was examined the altered protein levels are present in the lymph nodes of an NHL in vivo mouse model. Incorporating proteomic studies, we evaluated protein changes of two IA concentrations in an in vivo male Balb/c mouse model inoculated with U-937 cells compared to positive control methotrexate.

Differential protein expression from axillary or inguinal lymph nodes mass was analyzed for IA capability to decrease, inhibit lymphoma proliferation, induce apoptosis, or modify any molecular pathway through proteomics by mass spectrometry carried out by isobaric tags for relative quantitation (TMT) through ultra-high performance liquid chromatography-tandem mass spectrometry UPLC-MS/MS for parallel multiplexing experiments [15,16].

Additionally, IA molecular docking experiments were carried out using target 15 dysregulated proteins, comparing IA and MTX to determine in silico physicochemical, pharmacokinetic, and toxicological properties. Analyzing and searching through an enrichment analysis from relative quantitative comparisons of proteins allows us to understand their functional role in cellular fate through a biological system of an NHL experimental model.

## 2. Results

To evaluate changes in the proteome of the non-Hodgkin’s lymphoma experimental model against negative and positive controls as mentioned in methods (named C−, MTX), and the related anatomical location and IA concentration behavior, we compared based on IA doses (5 and 10 mg/kg determined from a previous work [13]), location of the lymphoma node mass development, stratifying the lymph nodes by grouping them into subgroups according to left axillary nodes (named 5LANM and 10LANM) and to right inguinal nodes (named 5RINM and 10RINM), and negative and positive controls, obtaining six groups. Since the symptoms begin upper or lower, with the growth of lymph nodes in the neck, upper chest, armpit, and then abdomen or groin, according to the Lugano staging system [17], six raw MS files with 20,453 peptides were analyzed and searched against mouse protein database to identify gene/protein names. Data generated were exported to an Excel file (Database Appendix A) containing the identified and quantified proteins for this project. In total, 2717 proteins were identified and quantified for this project searching against the mouse protein database. Proteins of relative quantitation were divided into two categories, fold change (FC) > 1.5 was considered up regulation, while FC < 0.67 (1/1.5) was considered as down regulation. The number of differentially expressed proteins obtained for each comparation is summarized in Table 1, observing the added number of changes into the right inguinal nodes regardless of IA dose.

Followed by an exhaustive bioinformatics analysis, we compared physical interactions, functional association, and genomic context (gene fusion, gene neighborhood, and phylogenetic profiles), even if they were not physically linked; from data across KEGG, Gene Ontology, and/or Reactome databases. Assessing the probability of the 2717 proteins via enrichment analysis to be in the same pathway, process, or related to any disease through a web interface and network visualization with 15 categories, we looked into Gene Ontology Molecular Function (GOMF), for example, chaperone binding and extracellular matrix structural constituent (exemplified in Figure 2); Biological Process (GOBP), for example, cytoplasmic translation, mRNA processing, or ribose phosphate metabolic processing (Appendix A); Cellular Component (GOCC), for example, contractile fiber, proteasome complex or mitochondrial protein complex (Appendix A); and the KEGG database, for example, carbon metabolism, endocytosis, oxidative phosphorylation, or ribosome (Appendix A).

An enrich profiler plot and list were generated to show the number of proteins to be in the same pathway, cellular process, or reported to diseases regarding each enrichment analysis GO (MF, BP, and CC), Reactome, and KEGG databases (Figure 3). A dashed cutoff at the top of the graph divides the most significant enriched cellular processes, and black circles with the numbers of different processes of interest are highlighted according to each database according to the order in which they appear in a general list of enriched processes, enlisting the representative processes marked and arranged by significance value, with the first being with the highest value, the list shows the number in the general gradient of the process and database with which the process was enriched; there is an identification of the process according to each platform, with the enriched process named according to each database and significance value of the process (shown with number and color scale and arranged in a descending order). MF was the most representative and BP has the greatest enrichment. Finding the greatest diversity in BP, followed by MF, CC, Reactome, and KEGG, relevant and interesting interactions were found among consulted sources corresponding to cytoplasm, protein binding, metabolic processes, mitochondrion, catabolic process, response to stress, cell death, apoptotic process, structural molecular activity, and glycolysis/gluconeogenesis, to mention the topmost relative to 2199 protein intersections displayed in the enrich profiler table, with at least 20 significantly enriched processes and 120 subprocesses (see Appendix A; to identify gene/name symbol in UNIPROT ID, may refer to Appendix A).

To analyze the processes of deregulated proteins among treatments and/or node mass location, we compared the single fold change of proteins down regulated and up regulated from comparisons between the negative control (C−) and the treatments (5LANM, 5RINM, 10LANM, 10RINM, and MTX). We estimated the number of protein interactions among themselves related to the enrichment type (GOMF, GOBP, GOCC, and KEGG) to be in the same pathway or process; that was more than what we would expect for a random set of proteins of the same size and degree of distribution drawn from the genome, indicating that they were biologically connected.

Firstly, we compared behaviors through data shown in Table 1, visualizing categories, size of interactions, and fold change from 412 deregulated proteins for treatments (enrich profiler plot Appendix A) and via network enrichment through GOMF (Figure 4), and all annotations referring to proteins are based on the UNIQID gene name or UNIPROT-ID. We show comparison of down-regulated (76) and up-regulated (69) proteins from C− versus 5LANM (Figure 4A), with altered proteins related to cytoskeleton organization, actin binding, or microfilament activity (e.g., down-regulated Actn3 or Myh1; up-regulated Marcks or S100a4), or nucleosome binding or chromatin DNA binding (e.g., up-regulated H1f1 or H2afy). We compare down-regulated (117) and up-regulated (72) proteins from C− versus 5RINM (Figure 4B) with altered proteins related to FATZ binding (e.g., down-regulated Actn2, Myoz1, or Myoz3), structural constituent cytoskeleton (e.g., down-regulated Myom1 or Mybpc2), or phosphotransferase activity (e.g., up-regulated Ckb). We compare down-regulated (78) and up-regulated (80) proteins from C− versus 10LANM (Figure 4C) with altered proteins related to cytoskeletal motor activity (e.g., down-regulated Myh4 or Myh8), or phosphatidylcholine-sterol O-acyltransferase activator activity (e.g., up-regulated Apoe, Apoa1, or Apoa4). We compare down-regulated (83) and up-regulated (132) proteins from C− versus 10RINM (Figure 4D) with altered proteins related to actin binding (e.g., down-regulated Tnnc2 or Myl3) and structural constituent of the ribosome (e.g., up-regulated Rplp1 or Mrpl11). Additionally, we compare down-regulated (111) and up-regulated (63) proteins from C− versus MTX (Figure 4E) with altered proteins related to proton transmembrane transporter activity (e.g., down-regulated Slc25a4 or Mtco2) or enzyme inhibitor activity (e.g., up-regulated Cd55 or Hexim1), with a shared process or changed proteins among treatments.

Comparison via network enrichment by GOBP (Appendix A), shows comparison of down/up-regulated proteins from C− versus 5LANM (Appendix A) with altered proteins related to the muscle system process (e.g., down-regulated Tnnc2, Myh4, Myom3, or Actn3) and sarcomere organization (e.g., down-regulated Ldb3 or up-regulated Csrp1). We show comparison of down/up-regulated proteins from C− versus 5RINM (Appendix A) with altered proteins related to actomyosin structure (e.g., down-regulated Acta1; or up-regulated Zyx or Pdlim1), cellular component assembly involved in morphogenesis (e.g., down-regulated Myoz1 o Tmod4), or non-membrane-bounded organelle assembly (e.g., up-regulated Mrto4, Nup62 or Chmp5). We show comparison of down/up-regulated proteins from C− versus 10LANM (Appendix A) with altered proteins related to myofibril assembly (e.g., down-regulated Flnc or Casq1) and muscle contraction (e.g., down-regulated Pgam2 or up-regulated Atp1a3). We show comparison of down/up-regulated proteins from C− versus 10RINM (Appendix A) with altered proteins related to cytoplasmic translation (e.g., up-regulated Rps13 or Rplp1), muscle cell development (e.g., down-regulated Cfl2 or up-regulated Wdr1), or the muscle system process (e.g., down-regulated Aldoa or Atp2a1). Additionally, we show comparison of down/up-regulated proteins from C− versus MTX (Appendix A) with altered proteins related to the ATP metabolic process (e.g., down-regulated Ak1, Aldoa, or Eno2) and generation of precursor metabolites energy (e.g., down-regulated Eno3 or up-regulated Ppp1cc). These all have shared processes or changed proteins among treatments.

Compared via network enrichment by GOCC (Appendix A), we show comparison of down/up-regulated proteins from C− versus 5LANM (Appendix A) with altered proteins related to contractile fiber (e.g., down-regulated Myl1 or Slc4a1; or up-regulated Csrp1) and myofilament (e.g., down-regulated Tpm2 or Tnnt3). We compare down/up-regulated proteins from C− versus 5RINM (Appendix A) with altered proteins related to actin filament bundle (e.g., down-regulated Tpm3 or Pdlim7; or up-regulated Zyx or Pdlim1) and myosin complex (e.g., down-regulated Mylpf or Myh3). We compare down/up-regulated proteins from C− versus 10LANM (Appendix A) with altered proteins related to striated muscle thin filament (e.g., down-regulated Ttn or Acta1) and Z disc (e.g., down-regulated Pygm or Fhl3). We compare down/up-regulated proteins from C− versus 10RINM (Appendix A) with altered proteins related to ribosome (e.g., down-regulated Isg15; or up-regulated Rpl34), I band (e.g., down-regulated Atp2a1; or up-regulated Pdlim3), and polysome (e.g., up-regulated Rpl24). Additionally, we compare down/up-regulated proteins from C− versus MTX (Appendix A) with altered proteins related to contractile fiber (e.g., up-regulated Tnni2 or Dbi) with shared processes or changed proteins among treatments.

Compared via network enrichment by KEGG (Appendix A), we show comparison of down/up-regulated proteins (using UNIPROT nomenclature/gene name, see Appendix A) from C− versus 5LANM (Appendix A) with altered proteins related to necroptosis (e.g., down-regulated Q9WUB3/Pygm or P27661/histone H2AX; or up-regulated Q9CQ10/Chmp3 or Q8BX10/Pgam5) and biosynthesis of amino acids (e.g., down-regulated P21550/Eno3, P17183/Eno2, or O70250/Pgam2; or up-regulated Q9DCC4/Pycrl). We compare down/up-regulated proteins from C− versus 5RINM (Appendix A) with altered proteins related to chemical carcinogenesis, reactive oxygen species (e.g., down-regulated P48962/Slc25a4, P51881/Slc25a5, Q9CQQ7/Atp5f1, or P12787/Cox5a), and prion disease (e.g., down-regulated Q05144/Rac2). We compare down/up-regulated proteins from C− versus 10LANM (Appendix A) with altered proteins related to arginine and proline metabolism (e.g., down-regulated Q6P8J7/Ckmt2 or P07310/Ckm; or up-regulated P29758/Oat) and systemic lupus erythematosus (e.g., up-regulated Q8CGP2/HIST1H2BL). We compare down/up-regulated proteins from C− versus 10RINM (Appendix A) with altered proteins related to neutrophil extracellular trap formation (e.g., up-regulated P62806/Hist1h4a) and coronavirus disease (e.g., up-regulated P47915/Rpl29 or P47963/Rpl13). We compare down/up-regulated proteins from C− versus MTX (Appendix A) with altered proteins related to oxidative phosphorylation (e.g., down-regulated Q9DC70/Ndufs7 or P00405/Mtco2) and thermogenesis (e.g., down-regulated P41216/Acsl1 or Q9CQ69/Uqcrq), with shared processes or changed proteins among treatments.

Since, for the 412 proteins identified, we observed shared and specific changes or variations in each treatment, we looked for how many proteins were shared or changed individually among the different conditions. Concerning to 5LANM, 10LANM, and MTX versus negative control, we identified 283 deregulated proteins in correlation with the number of intersections. Through Venn diagrams (Figure 5A) and fold change networks (Figure 5B), we detected 27 (9.54%) altered proteins specific for 5LANM, 5 down (Tpm3, Uqcrh, Nt5c3a, Pdlim3, or Hck) related, respectively, to cytoskeleton, oxidative phosphorylation, nucleotidase, cytoskeleton, and proliferation, and 22 up (e.g., Oxct1, Hist1h1d, Prcp, Sh2d1a or Hist1h1a) related, respectively, to ketone metabolism, chromatin remodeling, carboxypeptidase, lymphocytic activation, and chromatin remodeling; 45 (15.9%) proteins specific for 10LANM, 6 down (e.g., Tgfb1, Lama2, Tsc22d4, Mprip, or Zmat2), related, respectively, to growth and differentiation, migration and organization, transcriptional repressor, cytoskeleton, and DNA-binding, and 39 up (e.g., Anxa5, Ccdc90b, Hmgn5, Zyx, or Med8) related, respectively, to cytoskeleton, mitochondrion, cellular transcription, cytoskeleton, and transcription regulation; 84 (29.68%) proteins specific for MTX 46 down (e.g., Ldha, Ndufa9, Ddost, Cyc1, or Rps2) related, respectively, to oxidoreductase, respiratory chain, protein glycosylation, oxidative phosphorylation, and synthesis of proteins; and 38 up (e.g., Rbp4, Ttr, Mup3, Gapvd1, or Serpina3k) related, respectively, to cytoskeleton, hormone activity, cellular component, GTPase-activity, and protease inhibitor. For 5LANM and 10LANM 38 (13.43%) shared proteins (9 down e.g., Stat1, Pdlim5, Iigp1, Isg15, or Fxr2) related, respectively, to signal transducer and transcription activator, cytoskeleton, GTPase, protein tag, and mRNA-binding protein; and 29 up (e.g., Ltf, Polr2e, Pgam5, Casp7, or Hist1h2af) related, respectively, to regulator activity, transcription of DNA into RNA, phosphatase activity, programmed cell death, and transcription regulation. Additionally, 65 (22.97%) shared proteins among 5LANM, 10LANM, and MTX 59 down (e.g., Ndufb4, Pygm, Spg20, Myh1, or Myl3) are related, respectively, to respiratory chain, glycogen catabolism, cell differentiation, and both cytoskeleton; and 5 up (e.g., Gar1, Timd4, Rplp1, Phactr4, or Marcks) are related, respectively, to ribosome biogenesis, phosphatidylserine receptor, protein synthesis, catalytic activity, and cytoskeleton; see Appendix A, down regulated (Appendix A) and up regulated Appendix A.

Regarding to 5RINM, 10RINM, and MTX versus negative control, we identified 374 deregulated proteins in correlation with the number of intersections. Through Venn diagrams (Appendix A) and fold change networks (Appendix A), we detected altered 61 (16.31%) proteins individual for 5RINM (24 down Api5, Mybbp1a, Cox5a, Ipo5, or Lrrc59) related, respectively, to antiapoptotic factor, activate or repress transcription, oxidative phosphorylation, and both nuclear proteins; and 37 up (e.g., Krt10, Ruvbl2, Wdr1, Zyx, or Akap2) related, respectively, to structural molecule, catalytic activity, cytoskeleton, and signal transduction, and binds to regulatory subunit (RII); 117 (31.28%) proteins individual for 10RINM 17 down (e.g., Prosc, Ptgr1, Cmbl, Clybl, or Tbcc) related, respectively, to homeostatic regulation, catalytic activity, hydrolase activity, catalytic activity, and cytoskeleton; and 100 up (e.g., Rbm4b, Cd209b, Cpa3, Mrpl11, or Pgam5) related, respectively, to translational activation, antigen-like, catalytic activity, mitochondrial gene expression, and catalytic activity, and 54 (14.44%) protein individuals for MTX 18 down (e.g., Ndufs7, Camk2d, Uqcrq, Rps2, and Cox6c) related, respectively, to rRNA processing, and metabolic pathways, regulation of Ca^2+^ homeostasis, oxidative phosphorylation, synthesis of proteins, and oxidative phosphorylation, and 36 up (e.g., Lonp1, Gpalpp1, Lyve1, Stfa3, or Azgp1) related, respectively, to catalytic activity, transcription factor, transporter, hydrolase activity, and lipid degradation. For 5RINM and 10RINM 22 (5.88%), shared proteins 6 down (e.g., Mcm4, Isg15, Pdlim5, Nt5c3a, or Ing1) related, respectively, to DNA replication, ubiquitination, cytoskeleton, catalytic activity, and transcriptional activation, and 16 up (e.g., Sp3, H3f3a, Cfh, Alpl, or Cd48) related, respectively, to transcription factor, chromatin remodeling, cell adhesion, catalytic activity, and signaling. Additionally, 62 (16.58%) shared proteins concerning 5RINM, 10RINM, and MTX 56 down (e.g., Actn3, Tnnc2, Eno2, Eno3, or Atp2a1) related, respectively, to nucleosome binding, cytoskeleton, both to glycolysis, and catalytic activity, and 5 up (Cfd, Marcks, Phactr4, Tf, and Fetub) related, respectively, to catalytic activity, cytoskeleton, hydrolase activity, cell differentiation, and protease inhibitor. There are also display networks in Appendix A for down regulated (Appendix A) and for up regulated (Appendix A), and they are summarized in Appendix A. Concerning this comparison, we detected Rpl18 protein down in 5RINM and MTX, but up in 10RINM, plus Rpl7 and Rpl37 proteins up in 10RINM and down for MTX; these three proteins (ribosomal proteins) showed opposite behavior among treatments (Appendix A).

Through these comparative analyses, we obtain proteins that are exclusively down or up regulated in each treatment or group (Appendix A). For 5LANM, seven proteins, none down regulated, and seven up regulated (e.g., Sh2d1a, Ces1c, S100a9, Prcp, or Plbd1) related, respectively, to regulation of immune response, cell differentiation, programmed cell death, metabolic process, and phospholipase activity. For 10LANM, 29 proteins, 2 down regulated (Lama2, and Zmat2) related, respectively, to cell adhesion and gene expression, and 27 up regulated (e.g., Mug1, Oat, Hmgcl, Impdh1, or Gla) related, respectively, to protein binding, metabolic pathways, metabolism, growth progression, and catalytic activity. In the case of 5RINM 45 proteins, 17 down regulated (e.g., Rnf213, Api5, Hbb-b1, Cox5a, or Tuba4a) related, respectively, to angiogenesis, antiapoptotic factor, oxygen transport, cellular respiration, and cytoskeleton organization; and 28 up regulated (e.g., Krt10, Purb, Smarce1, Rnf13, or Crym) related, respectively, to microtubule cytoskeleton, gene transcription, transcriptional activation and repression, cell proliferation, and access to nuclear receptors. For 10RINM 78 proteins, 12 down regulated (e.g., Eci1, Vcp, Fbln1, Cmpk2, Cmbl, Clybl, Tbcc, Ptgr2, Ptgr1, Nipsnap3b, and Prosc) related, respectively, to catalytic activity, RHO GTPase cycle, extracellular matrix, biosynthesis of cofactors in mitochondria, and metabolic pathways; and 66 up regulated (e.g., Mocs3, Clta, Gfra2, Adh1, or Chtop) related, respectively, to gene expression, mitotic spindle, MAPK1/MAPK3 signaling, glycolysis/gluconeogenesis, and transport of mature transcript to cytoplasm. Additionally, for MTX 48 proteins, 18 down regulated (e.g., Ptprc, Anxa6, Rplp0, Ass1, and Ndufs7) related, respectively, to immune system, programmed cell death, translation, biosynthesis of amino acids, rRNA processing, and metabolic pathways; and 30 up regulated (e.g., Agrn, Gc, Ahsg, Dbi, or Eif4h) related, respectively, to ATPase regulator activity, cytoskeletal protein binding, metabolism of proteins, mitochondrial fatty acid beta-oxidation, and translation.

Complementarily, since we observed shared and specific changes and/or variations in each treatment, we compare across networks via enrichment analysis of the 412 up- or down-regulated proteins from C− versus 5LANM, 5RINM, 10LANM, 10RINM, and MTX, to GO (Figure 6) and KEGG displaying the altered biological processes based on the number of genes related (Appendix A) that are shared among the different conditions, such as nucleosomal DNA binding, biosynthesis of amino acids, transcription, translation, cytoskeletal structure, metabolic and energy process, glycolysis/gluconeogenesis, and cell death compromised through necroptosis. Additionally, the analyses show dysregulated proteins related to other disorders; for example, coronavirus disease—COVID-19, systemic lupus erythematosus, Huntington disease, or Parkinson disease. Correspondingly, the analyses with implicated proteins through GO (Appendix A) and KEGG identify distinctive molecules and cellular processes (Appendix A). Displaying for 5LANM Oxct1 (key enzyme for ketone body catabolism) and Chmp3 (endosomal sorting); for 10LANM Phb (signaling functions), Impdh1 (RNA and/or DNA metabolism); for 5RINM Cirbp (genotoxic stress response), Pdlim1 (cytoskeletal protein), Plg (proteolytic factor, leads to cell detachment and apoptosis); and for 10RINM H1f0 and Hist1h1b (H1f5) (condensation of nucleosome), and Baz1b (chromatin remodeling and acts as a transcription regulator).

Finally, we perform in silico molecular binding in a model pointing 15 dysregulated proteins, matching IA and MTX ligands with protein targets. In the case of IA, it has better affinity to fragile X mental retardation syndrome-related protein 2 (P51116), DNA-directed RNA polymerase II subunit E (A0A087WVZ9), cold-inducible RNA-binding protein (Q14011), zinc-alpha-2-glycoprotein (P25311), and caspase-3 (P42574), with ∆G values of −6.83, −5.18, −5.92, −6.08, and −5.29, respectively; also, they share several amino acid residues interactions as well as binding positions with the pharmacological MTX control. In addition, IA also showed better affinity to TSCC22 domain family protein 4 (Q9Y3Q8), apoptosis inhibitor 5 (Q9BZZ5), alpha-1-acid glycoprotein 1 (P02763), nuclear pore glycoprotein P62 (P37198), and caspase-7 (P55210), with ∆G values of −4.37, −5.19, −5.15, −4.07, and −6.12, respectively, in comparison with MTX, although they bind to a different position than MTX, possibly at a pharmacological allosteric site (Table 2, Figure 7).

The analysis of molecular docking of histone H2A type 1-F (Q8CGP5), 40S ribosomal protein S2 (P15880), signal transducer and activator of transcription 1 (P42224), transforming growth factor beta-1 protein (P01137), and beta-enolase (P13929) showed better affinity to MTX with ∆G values of −4.24, −7.02, −5.12, −4.72, and −5.83, respectively. Nevertheless, the ∆G values obtained of IA on these proteins were important (Table 2, Figure 7).

In silico servers were used to determine the ADMET process and the toxicological properties of both molecules. Cheminformatics analysis is currently considered an important tool in the development of new molecules that may be drug candidates. Taking this into account, the analysis was carried out, the results of which are shown in Table 3.

## 3. Discussion

As a result of analyzing the differential protein expression from axillary (LANM) and inguinal (RINM) lymph nodes masses, we noticed that IA treatment affects proteins related to chromatin remodeling and DNA replication in, for example, down regulation of the Mcm4 protein, a helicase that plays a central role in initiation and elongation of DNA replication [18], as well as up regulation of the Hmgn5 protein that binds to euchromatin and modulates cellular transcription by counteracting linker histone-mediated chromatin compaction [19]. Several histones, such as up-regulated Hist1h2af, a core component of nucleosome, that wrap and compact DNA into chromatin, permitting or limiting DNA accessibility to the cellular machineries, through post-translational modifications, play a central role in transcription regulation, DNA repair, DNA replication, and chromosomal stability [20].

This modification correlates to transcription and translation variation, with up regulation, for example, of Med8 a protein coactivator involved in the regulated transcription of nearly all RNA polymerase II-dependent genes and the general transcription factors [21]. Down regulation of Fxr2, an mRNA-binding protein that acts as a regulator of mRNAs translation and/or stability, leads to the accumulation of cytoplasmic Nup condensates and nuclear morphology defects [22,23]. Up regulation of Polr2e a DNA-dependent RNA polymerase catalyzes transcription, a common component of RNA polymerases I, II, and III, which synthesize ribosomal RNA precursors, mRNA precursors, and many functional non-coding RNAs, and small RNAs, such as 5S rRNA and tRNAs, respectively [21]. Additionally we found synthesis of ribosomal proteins, for example, down-regulated Rps2, or Mrps22, and up-regulated Rps23, Rps25, or Rplp1 [24,25].

Signal transduction is disturbed through proteins as down-regulated Stat1, a transcription activator that mediates cellular responses to interferons (IFNs), cytokine KITLG/SCF, and other cytokines and other growth factors. Following type I IFN (IFN-alpha and IFN-beta) binding to cell surface receptors, signaling via protein kinases leads to activation of Jak kinases (TYK2 and JAK1) and to tyrosine phosphorylation of STAT1 and STAT2. This may mediate cellular responses to activated FGFR1, FGFR2, FGFR3, or FGFR4. This also plays a key role in activation of cellular immunity, and subsequently, stimulation of antitumor immune response via IFN-gamma (pro-apoptotic and antiproliferative functions [26]).

The dysregulation also can be seen in growth, proliferation, or differentiation proteins; for example, for down regulation of Tgfb1 protein phosphatase that play a key role in balancing the cellular responses to the transforming growth factor-β (TGFβ) signals, its activity is essential in controlling SMAD3 protein levels [27]. Down regulation of Ing1, a transcriptional activation protein that participates in cellular proliferation, apoptosis, and cellular senesce, and whose upregulation inhibit cell growth and metastasis, of which isoform 1 may function as an oncoprotein, and isoform 2 acts as a negative growth regulator acting as a tumor suppressor [28,29], and is a reader of histone modification marks, promoting the deacetylation of the rRNA transcription factor (UBF) and inhibiting its activity through interaction with mTOR, which regulates the translation of mRNAs [30].

Another example, such as the Tsc22d4 protein down regulated, participates in cellular senescence suppression and metabolic regulation through its interaction with Akt1 trigger energy deprivation and oxidative stress; reduction in Tsc22d4 improves survival through the improvement of mitochondrial complex I gene expression [31,32]. The up-regulated Cirbp protein responds to genotoxic stress by stabilizing transcripts of genes involved in cell survival, operating as a translational activator, and also appears to play a crucial role in suppression of cell proliferation [33]. The up-regulated Azgp1 protein may serve as prognostic value, since it stimulates lipid degradation in adipocytes and causes the extensive fat losses associated with some advanced cancers [34,35]. A protein with intracellular signaling and regulation activity is the up-regulated Alpl, a homodimeric cell surface phosphohydrolase, that participates in tumorigenesis and cancer progression by decreasing migration and invasion and acting as a tumor suppressor [36,37] or as an oncogene [38]. Additionally, there is the up-regulated Orm1, at RINM, an alpha-1-acid glycoprotein 1 linked to tumor immunity that participates as a blocking factor to protect tumor cells against immunological attack, promoting the immune escape of tumors implicated with reduced overall survival and lymph node metastasis [39,40]. Another related is the mitochondrial DNA-binding up-regulated protein Ssbp1, which regulates mitochondrial and nuclear H_2_O_2_ levels [41].

Reactive oxygen species (ROS) are important in regulating normal cellular processes, although are apparently influenced in contradictory ways, either stimulating tumorigenesis and supporting transformation of cancer cells or causing cell death [42]. Oxidative stress recently has been involved as a characteristic that might induce the proliferation and differentiation of tumor cells and promote tumor progression, showing that key genes involved in oxidative stress could predict prognosis and potentially become therapeutic targets [43], providing information on how signaling pathways could be manipulated during the treatment of neoplasms [44].

Correspondingly, many cytoskeletal proteins were found dysregulated; for example, up regulation of Zyx, an adhesion protein that binds to alpha-actinin and CRP protein, is important for the targeting of members of focal adhesions and the formation of actin-rich structures [45]. Down-regulated Pdlim5, or up-regulated Pdlim1, act as adapters that bring other proteins (such as kinases) to the cytoskeleton, which is involved in assembly, disassembly, and addressing fibers (e.g., ACTN1 or PALLD), as well as cell migration and maintaining cell polarity [46]. Down-regulated Pdlim3 is critical for sonic hedgehog signal transduction, as its elimination represses the tumor growth [47] and metastasis [48]. We found that up-regulated proteins that imply the nuclear pore complex (NPC) disassembly; for example, nucleoporins Nup35, Nup62, and Nup54. Nup35 is part of the inner ring and essential for the formation of the NPC and nuclear envelope (NE), binds to the nuclear membrane, and induces membrane deformation to facilitate the formation of NPC during the interphase, allowing the insertion of nucleoporins in NE [49]. Nup62 has an essential role in nuclear transportation, cell migration, and cell cycle regulation, interactions between nucleoporin and chromatin stimulate the cell-cycle gene expression inside the nucleoplasm. Nup62 cooperates with Nup88, which interacts with NF-κB, promoting proliferation and cell growth, so Nup88 is stabilized through overexpression of Nup62 [50,51,52]. Overexpression of Nup62 promotes colony formation and cell migration in cancer gastric [53].

Related to the metabolic and energy process, for oxidative phosphorylation, we found down-regulated Aldoa, Ldha, Eno2, or Eno3 involved in glycolysis with catalytic activity on behalf of the generation of precursor metabolites and energy [54]. Bonding with inhibition of IA was shown in vitro of glycolytic enzymes (Ldha and Aldoa) [13]. Additionally, we found molecules related to the respiratory chain; for example, down-regulated Ndufa9 or Cyc1, part of the oxidative phosphorylation system (OXPHOS) complexes in the inner mitochondrial membrane that is an intricate process, with catalytic activities, and a large number of subunits that play essential roles in assembly, regulation, and stability [55]. For glycogen catabolism, down regulation of Pygm protein plays a central role for cellular maintenance and organismal glucose homeostasis, an allosteric enzyme that catalyzes the rate-limiting step in glycogen catabolism, and the phosphorolytic cleavage of glycogen to produce glucose-1-phosphat; this change is critical to prevent tumors as a treatment-effective option through inhibiting tumor growth in various ways [56]. Such is the case. With other metabolic enzymes, it probably stimulates the tricarboxylic acid cycle and oxidative phosphorylation, which could result in enhanced ROS production [57]. Some NHL cases show double expression of c-Myc and BCL2, or BCL6; or triple expression, being more aggressive neoplasms [58]. These proteins also have been implicated in common cytogenetic abnormalities, Bcl-2 inhibits pro-apoptotic proteins and is associated with poor outcomes, and its expression significance remains under investigation. Alford, et al., in 2024, found no significant differences between Bcl-2-negative patients compared with patients who expressed high Bcl-2, suggesting that other prognostic markers may have a more pronounced impact [59]. Accordingly, in the present trials, we did not find changes in its expression relative to the treatments, which may be due to the pathways in which IA has an effect.

As we perceived previously, in vitro IA might interact with cytoskeleton proteins as their targets and promote an antitumor effect or cell death by energy metabolism disruption [13]. With different activities through treatments, for example, factor H, a protein up regulated in RINM but not in LANM, a decay accelerator of the alternative pathway convertases C3, high levels of Cfh decrease the risk of death [60,61]. Another up-regulated Pgam5 protein displays phosphatase activity for serine/threonine residues, acting as a central mediator for programmed necrosis induced by TNF, via reactive oxygen species and calcium ionophore [62]. Another form of regulated cell death that is initiated by disturbances of extracellular or intracellular homeostasis is the necroptosis, which triggers a specific necrotic cell death pathway involving activation of RIPK3 and MLKL via ZBP1, with an important role in tumorigenesis that implies the potential of targeting necroptosis as a cancer therapy. Its molecular mechanism is well known, with recent studies of the regulation and function. The ZBP1 protein was up regulated in 10RINM, which acts as an essential mediator of pyroptosis, necroptosis, and apoptosis (PANoptosis), an integral part by activating RIPK3 kinase, which phosphorylates and activates MLKL (characterized by calcium influx and plasma membrane damage), caspase-8 (CASP8), and the NLRP3 inflammasome. Additionally found is caspases Casp3 up regulated in 5RINM; and Casp7 up regulated in 10RINM, 5LANM, and 10LANM, and thiol proteases involved in different programmed cell death processes, such as apoptosis [63,64], or Api5, an antiapoptotic factor that negatively regulates ACIN1 by binding to it, suppressing cleavage from CASP3, and it is also known to efficiently suppress E2F1-induced apoptosis [65].

The necroptosis process triggers disruption of the nuclear envelope and leakage of cellular DNA into the cytosol by initiating a pronecrotic kinase cascade, and triggers downstream reactive oxygen species production. Emphasizing the complex mechanisms that incline the equilibrium concerning different cell fates, dying in different ways, therefore, the death can be predetermined, programmed, and cleanly executed, as in the case of apoptosis, or it can be traumatic, inflammatory, as probably IA induces a type of necrosis by disrupting the homoeostasis, maybe related to mitotic catastrophe or anoikis [66,67,68,69]. Bhatti et al., 2017, find a synergistic combination BV6/Bortezomib that induces cell death, which is effective even when apoptosis is blocked, with important implications for the development of new treatment strategies for B-cell NHL, since it primarily triggers necroptosis or, alternatively, engages both apoptotic and necroptotic cell death [70]. Koch et al., 2021, looking for novel treatment strategies to improve the clinical outcome in patients with Burkitt’s lymphoma (a highly aggressive form of B-cell NHL), demonstrate that the combinational treatment with the Smac mimetic BV6 and TRAIL triggers necroptosis [71].

Our experimental in vivo model has preventive characteristics, since we inoculated the NHL cells and then after 24 h administered the treatment; therefore, it is as if we are taking a photograph of the events. It can be appreciated that there are changes at different pathways or processes as in cytoskeleton organization, transcription, translation, chromatin remodeling, ribosomal function, energy metabolism, glycolysis/gluconeogenesis, cell proliferation, or cell death compromised through necroptosis or apoptosis. With subsequent alterations in subprocesses or pathway derivations according to the involved genes, either shared or specifically in the analyzed treatment 5 mg/kg IA, 10 mg/kg IA, or 1.25 mg/kg MTX. These results confirm that IA has a dose-dependent effect previously observed to trigger changes in different processes that define the cellular fate [14], with differences regarding the MTX effect. NHL represents a broad spectrum of diseases that are some of the main causes of morbidity and mortality worldwide. Chemotherapy is an effective treatment against NHL, used alone or in combination with methotrexate, cyclophosphamide, and doxorubicin. However, to date, all the drugs used induce various side effects, as is the case with MTX, which is the drug of choice for the treatment of NHL in Mexico [7,8,9].

To reinforce our findings to understand the molecular effect of IA as treatment on NHL mechanisms with the docking experiments, we noticed that even IA binds to a different position regarding MTX on the same protein, and this could possibly be the allosteric sites with pharmacological interest. Additionally, when binding shows better affinity to MTX on ∆G values, the obtained ∆G values of IA on same proteins was important and we cannot discard it as a possible target to IA. We also noticed that this matches the molecular interactions with the expression levels, for example, in down regulation of apoptosis inhibitor 5 (Api5), an inhibitor apoptosis that plays a role in maintaining homeostasis, that usually is upregulated in cancers, Api5 is shown to regulate both apoptosis and cell proliferation [65].

In silico physicochemical, pharmacokinetic, and toxicological properties were determined to compare IA and MTX. As we can see in Table 3, both molecules comply with Lipinski’s rule of five [72,73] and Weber considerations, parameters that brings information about the disponibility of drugs after oral administration; in this sense, it is probably that both molecules must have good orally disponibility [74,75]. Additionally, the PAINS filter did not generate any alert on either molecule, as this parameter considers promiscuous molecules that can show biological activity over several pharmacological targets [76]. In respect to ADMET pharmacokinetics properties, IA showed better absorption than MTX, which is an important value to consider in determining the biodisponibility of molecules. The distribution prediction showed a higher volume of distribution and protein plasma binding to IA in comparison to MTX. The predicted metabolism of IA and MTX suggests that they are substrate only of CYP3A4 and does not inhibit any of the CYP most evaluated on informatic assays. Additionally, the excretion predicted clearance values of 4.64 and 2.41 mL/min/Kg with a half-life (t 1/2) of 0.78 and 0.39 h for IA and MTX, respectively. The ADME results obtained in our study are similar to the those reported in other in silico and in vivo studies [77]; moreover, our study provides valuable information about IA, and their comparison with MTX, an anticancer drug used in chemotherapies [6,8]; ADME information for molecules is an important parameter that must be determined, and considering the recently informatic advance, in silico techniques have been transformed as a valuable tool that allows us to determine these parameters, reducing costs, time, and avoiding the use of animals [78].

Subsequently, toxicity prediction brings us an important parameter, due to the results suggesting that IA does not generate hepatotoxicity and neurotoxicity in comparison to MTX, which can induce this kind of toxicity. Additionally, the results predict a human lethal dose 50 (LD_50_) or higher to IA of 1330 mg/kg in comparison to 3 mg/kg for MTX; this result suggests a wide therapeutic window for the use of IA in comparison to MTX. Finally, the results show that IA belongs to Class IV (300 < LD_50_ ≤ 2000 mg/kg), indicating that it produces low toxicity according to the Globally Harmonized System of Classification and Labeling of Chemicals [79,80,81]; on the other hand, MTX belongs to Class I, which is considered fatal by ingestion (LD_50_ ≤ 5), and considering all in silico results shown above, IA is expected to be quite safe for use as a drug.

The NHL is stratified based on the location of the lymphoma with respect to the diaphragm and its spread to organs; this makes it possible to identify the location of the disease and its extension. The Lugano staging system identifies four stages with Roman numerals I to IV depending on the presence of lymphoma in different areas of the body. Stage I involves a single lymph node region (e.g., cervical, axillary, inguinal, and mediastinal) or lymphoid structure, such as the spleen, tonsils, thymus, or Waldeyer’s ring. Stage II indicates that the lymphoma is in two or more node regions or lymph node structures on the same side of the diaphragm (upper or lower), considered to be “lateralized”. Stage III indicates lymph node regions or lymphoid structures on both sides of the diaphragm and there is lymphoma invasion into the spleen or splenic hilar, celiac, portal nodes, paraaortic, iliac, inguinal, or mesenteric nodes. Stage IV indicates that the lymphoma spread to extranodal organs or tissue outside the lymphatic system, such as the liver, lungs, or bone marrow [17].

Based on preceding classification, we stratified the node mass obtained from the mice treated with IA to distinguish the influence of the dose used and the anatomical location, probably related to LANM as stage II and RINM as stage III, according to the staging system. We managed to discern on this in vivo model that the dose plus the anatomical location of the node mass also influence the treatment response, triggering different shared or exclusive pathways that describe the tumor outcome. Currently, the stratification system is universally applicable, facilitating the assignment of treatment and comparison of results between patients. NHL treatment begins once the diagnosis is obtained, and the strategy to be followed is defined to eliminate the tumor cells. Chemotherapeutic schemes that are currently used involve the use of drugs that possess different activities in cells whose goal is to generate an amount of damage in the genomic, proteomic, or structural integrity, leading to the activation of cell death signaling [82]. Our findings at the proteomic level could provide information through future studies, with benefits in diagnosis, treatment, or patient prognosis, which is crucial for those suffering from NHL.

## 4. Materials and Methods

### 4.1. Incomptine A (IA) Isolation

Compound IA was isolated from the aerial parts of *Decachaeta incompta* (syn.: *Eupatorium incomptum*; Asteraceae) collected in the state of Oaxaca, Mexico. The plant was identified by the MS Abigail Aguilar Contreras taxonomist of the Instituto Mexicano del Seguro Social (IMSS). A voucher specimen (15311) was deposited at the Herbarium, Herbario Medicinal de México IMSSM. The extraction and isolation procedure were performed according to a protocol previously described [13]. Identification of IA was made by comparison (NMR, and HPLC-DAD), with an authentic sample having a purity near to 99%). Dimethyl sulfoxide (DMSO) was used to dissolve IA and methotrexate (PISA pharmaceutical). Molecule structures were generated with bio model software https://biomodel.uah.es/en/DIY/JSME/draw.es.htm (access 26 November 2024).

### 4.2. Chemicals and Instrumentation

The TMT6plex Isobaric Label Reagent Set, Pierce Quantitative Colorimetric Peptide Assay (Thermo Fisher Scientific, Waltham, MA, USA) was used. Triethylammonium bicarbonate buffer (1.0 M, pH 8.5 ± 0.1), Tris (2-carboxyethyl) phosphine hydrochloride solution (0.5 M, pH 7.0), iodoacetamide (IAA), formic acid (FA), acetonitrile (ACN), and methanol (Sigma, Burlington, MA, USA) were used. Trypsin from bovine pancreas (Promega, Madison, WI, USA) was used. Ultrapure water prepared from a Millipore purification system was used. An Ultimate 3000 nano UHPLC system (Thermo Scientific) coupled online to a Q Exactive HF mass spectrometer (Thermo Scientific) equipped with a Nano spray Flex Ion Source (Thermo Scientific), TMT-based Quantification Analytical Service from Creative Proteomics (New York, NY, USA) [83,84] was used.

### 4.3. Cell Culture Conditions

NHL cell line U-937 (histiocytic lymphoma, cat. CRL-1593.2) was obtained from the American Type Culture Collection (ATCC, Manassas, VA, USA). To develop the in vivo mice model, cells were cultured in RPMI 1640 culture medium (GIBCO, Waltham, MA, USA; Cat: 11875-093), penicillin (100 U/mL)/streptomycin (100 μg/mL), and added with 5% fetal bovine serum (GIBCO Cat: 16000044) in a 37 °C incubator and 5% CO2. Cell cultures were maintained at a density of 2.5 × 10^6^ cells in T75 flasks (Invitrogen, Waltham, MA, USA) [85].

### 4.4. Animals

To induce the lymphoma in vivo model, we use healthy male Balb/c mice (25 ± 3g), provided by the animal house from the Centro Médico Nacional S-XXI, of the Instituto Mexicano del Seguro Social (IMSS). The research was conducted with ethical authorization by the National Committee of Scientific Research, IMSS (CNIC approval R-2018-785-111). Animals were maintained under controlled conditions, with temperature (at 22 °C ± 2 °C), humidity, and light–dark periods with a cycle of 12 h, with ad libitum access to food and water, in clean and sterile polyvinyl cages. Concurrent to the technical specifications for the production, care, and use of laboratory animals of the Mexican Official Norm, NOM-062-ZOO-1999 [86].

#### 4.4.1. Anti-Lymphoma Treatments

Considering a significant antitumoral effect previously determined due to the concentration that kills 50% cell population CC_50_ (cytotoxic activity) were in the NHL cell line U-937 after 24 h upon exposure to IA equivalent to 0.12 µM and 0.5 µM for MTX as a drug control, and that the apoptosis activation was shown by DAPI staining attributed to the decrease in the percentage of viable cells, since the CC_50_ was ≤30 μg/mL with antitumoral potential, according to the guidelines established by the program for the development of natural products with antineoplastic activity of the National Institute of Cancer [13,87], to evaluated the activity, we use IA doses of 5 mg/kg and 10 mg/kg, and MTX 1.25 mg/kg as drug control, 24 h after cell inoculation of the cell line U-937 in the anti-lymphoma mice model [13].

#### 4.4.2. In Vivo Lymphoma Male Balb/c Mice Model

To induce the in vivo lymphoma model, 24 male Balb/c mice were inoculated intraperitoneally with 5 × 10^5^ thousand/100 μL U-937 histiocytic lymphoma cells each (type of non-Hodgkin’s lymphoma) [88,89,90] and divided into four random groups (six animals per group). After, 24 h of cell inoculation treatments were administered orally during eight days as follows: negative control (C−) treated with 0.5 mL vehicle (DMSO, 1% *v*/*v* in PBS), positive control (MTX) methotrexate 1.25 mg/kg, and IA treatments of 5 mg/kg and 10 mg/kg. The animals were monitored for 30 days, tracking weekly weight, survival, and metabolic test (comprising 1 mL water intragastric administration, and collecting urine and feces in a metabolic chamber for 2 h) to identify pathologic alterations related to progression of disease or toxic effects. Subsequently, to complete observations, all animals were euthanized by cervical dislocation, and axillary and inguinal lymph nodes were removed, measured, and weighed. They were then preserved in saline solution at −70 °C until use [91,92].

### 4.5. Non-Hodgkin’s Lymphoma Protein Expression Induced Through IA

To evaluate the changes in the proteome of the non-Hodgkin’s lymphoma experimental model, we clustered the lymph nodes as pools of all mice in each group, concerning IA treatment 5 mg/kg and 10 mg/kg, and in order to stratify the location of the disease due to lymphoma development, lymph nodes were gathered in subgroups according to left or right axillary nodes, and to left or right inguinal nodes, obtaining six lymph nodes pools as follows: (C−) negative control treated with vehicle; (MTX) drug control methotrexate; (5LANM) 5 mg/kg IA left axillary node mass; (5RINM) 5 mg/kg IA right inguinal node mass; (10LINM) 10 mg/kg IA left axillary node mass; and (10RINM) 10 mg/kg IA right inguinal node mass [13].

#### Sample Preparation for TMT-Based Proteomic Analysis

The following are the next steps each lymph node pool: Protein extraction: Add an appropriate amount of tissue lysis buffer and lyse the tissues by using TissueLyser. Centrifuge at 12,000 rpm for 15 min at a low temperature and transfer the supernatant to a new EP tube. Determine the protein concentration by using a BCA kit. Protein digestion: Transfer 100 μg protein per sample into a new microcentrifuge tube. For each sample tube, reduce by 10 mM TCEP at 56 °C for 1 h. Alkylate by 20 mM IAA at room temperature in dark for 1h. Add free trypsin into the protein solution at a ratio of 1:50, and incubate the solution at 37 °C overnight. Lyophilize the extracted peptides to near dryness. Re-dissolve the sample with 100 mM TEAB. Peptide labeling: Immediately before use, equilibrate the TMT label reagents to room temperature. Add 41 μL of anhydrous acetonitrile to each tube. Allow the reagent to dissolve for 5 min with an occasional vortex. Briefly centrifuge the tube to gather the solution. Transfer the samples to the TMT reagent vial as sample/label reagent: (C−)/TMT6-126; (MTX)/TMT6-127; (5LANM)/TMT6-130; (5RINM)/TMT6-128; (10LINM)/TMT6-131; and (10RINM)/TMT6-129. Incubate the reaction for 1 h at room temperature. Add 8 μL of 5% hydroxylamine to the sample and incubate for 15 min to quench the reaction. Combine samples at equal amounts in a new microcentrifuge tube. Fractionation of the sample: fractionate of the labeled peptides with six components using HPLC [93].

### 4.6. Nano LC-MS/MS Analyses

For nanoflow UPLC, material and processes: Use a nanocolumn, trapping column (PepMap C18, 100Å, 100 μm × 2 cm, 5 μm), and an analytical column (PepMap C18, 100Å, 75 μm × 50 cm, 2 μm); loaded sample volume: 2 μg mobile phase: A: 0.1% formic acid in water; B: 0.1% formic acid in 80% acetonitrile. Total flow rate: 250 nL/min LC linear gradient: from 5 to 7% buffer B in 2 min, from 7% to 20% buffer B in 80 min, from 20% to 40% buffer B in 35 min, then from 40% to 90% buffer B in 4 min. Mass spectrometry: For TMT-labeled samples, the full scan was performed between 350 and 1650 m/z at the resolution 120,000 at 200 Th, and the automatic gain control target for the full scan was set to 3 × 10^6^. The MS/MS scan was operated in top 15 mode using the following settings: resolution 30,000 at 200 Th; automatic gain control target 1 × 10^5^; normalized collision energy at 32%; isolation window of 1.2 Th; charge sate exclusion: unassigned, 1, >6; and dynamic exclusion 40 s [94].

### 4.7. Data Analysis

The 6 raw MS files were analyzed and searched against a mouse protein database based on the species of the samples using Maxquant (v2.6.7.0), Proteome Discoverer 3.2 (ThermoFisher) and ProteoWizard (version 3). The parameters were set as follows: the protein modifications were carbamidomethylation (C) (fixed), oxidation (M) (variable), TMT-6Plex; the enzyme specificity was set to trypsin; the maximum missed cleavages were set to 2; the precursor ion mass tolerance was set to 10 ppm, and MS/MS tolerance was 0.6 Da. The following statistics analyses were carries out: Distribution of all the proteins identified according to the protein mass (kDa); distribution of 20,453 peptides identified according to the length; and distribution of 2717 the proteins identified according to sequence coverage. The determinations and data analytical report were performed by the Analytical Service from Creative Proteomics (New York, NY, USA) with *p*-values smaller than 0.05 in the analysis (where *p*-value <  0.05 indicates > 95% confidence of a change in protein concentration irrespective of the magnitude of the change) selected to designate differentially expressed proteins [95,96,97].

### 4.8. Differential Protein Analysis

Generated data were exported to an Excel file (Database Appendix A) containing a total of 2717 identified and quantified proteins for this project. Normalized TMT reporter values (C−)/TMT6-126 and (MTX)/TMT6-127 were used to determine protein differential expression between treatments (MTX)/TMT6-127; (5LANM)/TMT6-130; (5RINM)/TMT6-128; (10LINM)/TMT6-131; and (10RINM)/TMT6-129, establishing the following ratios MTX/C, and IA/C, IA/MTX. The most common fold change cutoffs at protein level accepted with a biological meaning are defined by 1.5 times up or down [98]; sometimes, there is less rigor for a fold change cutoff ratio of >1.32 or  <0.68 [99] or with fold change >1.2 or <0.83 in abundance [100]. Proteins of relative quantitation were divided into two categories. Fold change (FC) > 1.5 was considered up regulation while FC < 0.67 (1/1.5) was considered as down regulation. The number of differentially expressed proteins is summarized in Table 1, and each comparison was analyzed by means of cell signaling pathways and enrichment processes analysis.

### 4.9. Bioinformatic Methodology

The raw data or the fold change cutoff scale from >1.5 to <0.67 in continuous values cannot be used in the analysis process due the software requirements, since ratios are not symmetric around one. Therefore, the log ratios (log1.5) are the log-of-the-fold changes, i.e., log1.5 (condition1/condition2). Log ratios are used/plotted in graphs for the exploratory analysis of pathways and functional enrichment of cellular processes in each of the different treatments, generating a log ratio values matrix with each protein, as those are nicer to show because they center around 0, giving reductions a negative value and increments a positive value, and log ratios are symmetric around zero [97,101]. The log ratio data matrix was used to feed the algorithms for the analysis of cellular process enrichment networks and signaling pathways through programming language R v4.2.2 and Rstudio v3.1.4, applying the clusterProfiler v4.9.0, MSigDB in R (Molecular Signatures Database, Mouse MSigDB v2024.1.Mm), enrich plot and ggplo2 packages; using symbol and mouse ID data (org.Mm.eg.db), as well as the Kyoto Encyclopedia of Genes and Genomes (KEGG); Gene Ontology (GO) databases Biological Process (BP), Molecular Function (MF), and Cellular Component (CC); and Reactome. The enrichKEGG, enrichGO, and compareCluster functions were applied as indicated in the developer’s manuals. The results are plotted with the functions “emapplot”, “aplot_list”, “dotplot”, and “cnetplot” to visualization [102].

### 4.10. In Silico Studies, Molecular Docking

The chemical structure of ligand methotrexate (CID: 126941) was retrieved from the chemical library PubChem (https://pubchem.ncbi.nlm.nih.gov/, accessed on 10 September 2023), and this was optimized and submitted to energetic and geometrical minimization using Avogadro software (Avogadro: an open-source molecular builder and visualization tool. Version 1.2.0) [103]. The chemical structure of ligand incomptine A was drawn, optimized, and submitted to energetic and geometrical minimization using Avogadro software. The proteins histone H2A type 1-F (Q8CGP), fragile X mental retardation syndrome-related protein 2 (P51116), DNA-directed RNA polymerase II subunit E (A0A087WVZ9), 40S ribosomal protein S2 (P15880), signal transductor and activator of transcription 1 (P42224), transforming growth factor beta-1 protein (P01137), TSCC22 domain family protein 4 (Q9Y3Q8), apoptosis inhibitor 5 (Q9BZZ5), cold-inducible RNA-binding protein (Q14011), zinc-alpha-2-glycoprotein (P25311), alpha-1-acid glycoprotein 1 (P02763), nuclear pore glycoprotein P62 (P37198), beta-enolase (P13929), caspase-3 (P42574), and caspase-7 (P55210) were used as a target of the study. These were retrieved from the Alphafold database (https://www.alphafold.ebi.ac.uk/, accessed on 14 October 2024). Proteins were optimized, thus, total molecules of water and ions not needed for catalytic activity were stripped to preserve the entire protein structure. Polar hydrogen atoms were added, and proteins were ionized in a basic environment (pH = 7.4). Gasteiger charges were assigned; the computed output topologies from the previous steps were used as input files to docking simulations [103,104].

The molecular docking experiments were carried out using Autodock 4.2 software [105]; the search parameters were as follows: a grid-base procedure was employed to generate the affinity maps delimiting a grid box of 126 × 126 × 126 Å^3^ in each space coordinate, and with a grid points spacing of 0.375 Å, the Lamarckian genetic algorithm was employed as a scoring function with a randomized initial population of 100 individuals and a maximum number of energy evaluations of 1 × 10^7^ cycles, and the analysis of the interactions in the enzyme/inhibitor complex was visualized with PyMOL software (The PyMOL Molecular Graphics System, Ver 2.0, Schrödinger, LLC, DeLano Scientific, San Carlos, CA, USA) [105,106].

#### In Silico Physicochemical, Pharmacokinetic and Toxicological Properties

Pan-assay interfering compounds (PAINS) are useful for medicinal chemists to optimize compounds that may benefit our health, such as IA, making it helpful to know the properties related to absorption, distribution, metabolism, excretion, and toxicity (ADMET) so medicinal chemists may assess the effects or risks of IA on the human body. To determine the physicochemical, pharmacokinetic and toxicological properties, the ADMETlab [107], SwissADME [108], admetSAR [109], and PROTOX [110] servers were used. For this, IA was drawn on ChemDraw software (version 22.0.0.22), and then the SMILES code was obtained, and in the case of MTX, the SMILES code was obtained from PubChem library (CID 126941); in all servers, SMILES code was used to predict the properties.

### 4.11. Comparison of Shared Processes and Molecules

Proteins with differential changes between treatments were explored with the help of the G:Profiler tool available online; graphs and pathway enrichment tables were generated by comparing with KEGG, GO and Reactome databases [111]. Enrichment plot images generated in the *X*-axis show the number of different enriched processes according to each database, while the *Y*-axis plots the significance value obtained in ascending order (−log10 adj-*p*-value), a dashed cutoff at the top of the graph divides the most significant enriched cellular processes, and black circles with a number different processes of interest are highlighted according to each database, according to the order in which they appear in a general list of enriched processes. At the bottom is the list of the representative processes marked arranged by significance value, the first being the one with the highest value. The list is labeled by ID (number in the general list of the process), resource (database with which the process was enriched), term ID (identification of the process according to each platform), term name (name of the enriched process according to each database), and adj-*p*-value (significance value of the process) shown with number and color scale and arranged in a descending order. Finally, to know how many molecules and cellular processes are shared between the different conditions, the modulated proteins data and their change values were used to analyze them with the DiVenn v2.0 online tool as indicated by the developer [112], as well as the generation of Venn diagrams of proteins list using the Venny v2.1 platform following the instructions available in the manual tool [113]. All annotations to refer proteins are based on the UNIQID gene name or UNIPROT-ID. http://geneontology.org/; https://reactome.org/; https://www.uniprot.org/; or https://www.genome.jp/kegg/pathway.html (all accessed 26 November 2024).

## 5. Conclusions

Results from differentially expressed protein levels examined in lymph nodes of an in vivo mouse model confirm the previously observed dose-dependent effect of IA as a secondary metabolite with important potential as an anticancer agent for the treatment of NHL, showing that drug type or anatomical location influences treatment response by triggering modifications of pathways or processes, exclusive or shared between treatments, such as chromatin remodeling, transcription, translation, energy metabolism, oxidative phosphorylation, glycolysis/gluconeogenesis, cell proliferation, cytoskeletal organization, cell death compromised by necroptosis, and apoptosis, which define cell fate.

Although methotrexate is an important drug used in cancer treatment, IA is a good alternative which, based on in silico analysis, is likely to have better activity on different cancer-related targets, has better solubility, lower toxicity effects, a higher LD_50_, which translates into a wider range of doses to be used and, in addition to its pharmacological effect, is a good candidate for the development of a new anticancer drug.

Further research is needed to increase understanding of the role of IA as an antitumor drug to improve outcomes and survival of NHL patients, and to understand how it will affect pathways leading to cell death proteins. IA promises to be a potentially safer and more effective treatment to improve outcomes, reduce toxicities, and enhance survival in patients with NHL, initially targeting histones and transcription factors that will affect cell death proteins.

### Strengths and Limitations

Tandem Mass Tags (TMT) is a labeling technique used in quantitative proteomics, applicable to global proteomics and targeted analysis of post-translational modifications, involving the use of isobaric chemical tags (designed to be resistant to fragmentation) that consist of a mass reporter region, a mass normalization region, and a peptide reactive group. The intensity of these reporter ions reflects the relative abundance of the respective peptides in the samples, combined and analyzed together. The main advantages of the TMT method are its expanded multiplexing capability, allowing for simultaneous analysis of different samples in one run. This reduces run-to-run variability, increases throughput, and improves the statistical power of the analysis, which can result in a more accurate quantification. TMT also has its challenges. As ratio compression due to co-isolation and co-fragmentation of peptides can be an issue, potentially leading to inaccurate quantification. TMT is also more expensive than other quantitative proteomics techniques, such as label-free quantitation, which may limit its application in some cases. Furthermore, the complexity and large amount of data generated by TMT analysis require sophisticated bioinformatics tools and resources for accurate interpretation and validation of results. Another challenge around in silico prediction is the application domain or validation techniques of the models. These lead us to new research such as computational systems toxicology or data integration, which could be used in drug discovery such as IA, evaluating risks and usefulness so that doctors have compounds against NHL at their disposal.

## Figures and Tables

**Figure 1 pharmaceuticals-18-00242-f001:**
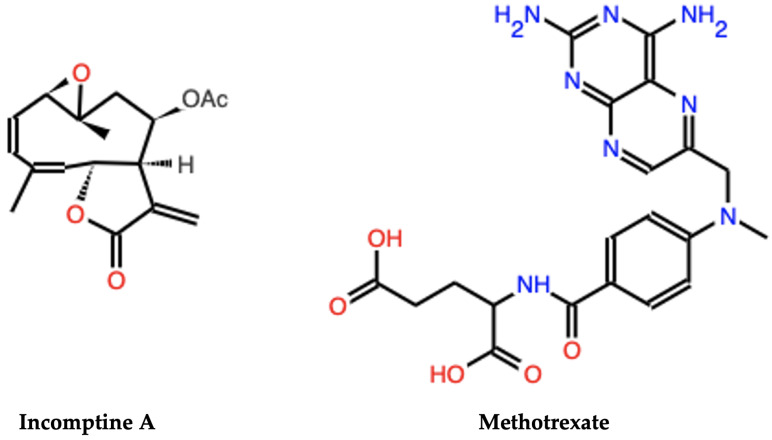
Structure of incomptine A (IA) and methotrexate (MTX).

**Figure 2 pharmaceuticals-18-00242-f002:**
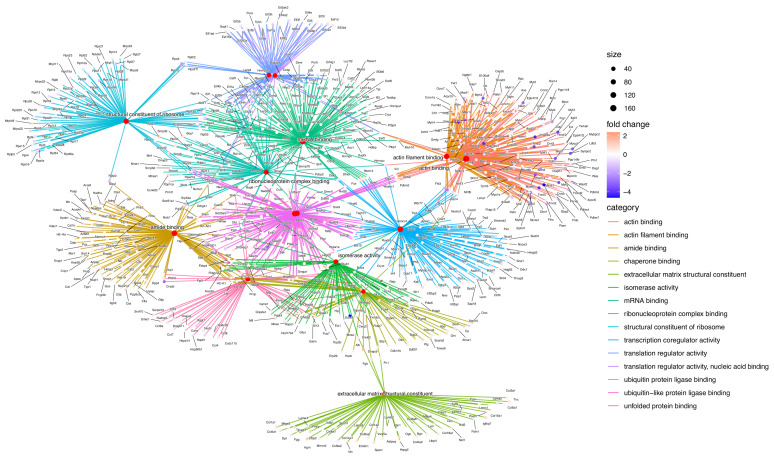
Network visualization from the 2717 proteins in a computational representation through Gene Ontology with enrichment analysis of molecular function subontology. It shows identified categories, and node size in range 40–160 interactions.

**Figure 3 pharmaceuticals-18-00242-f003:**
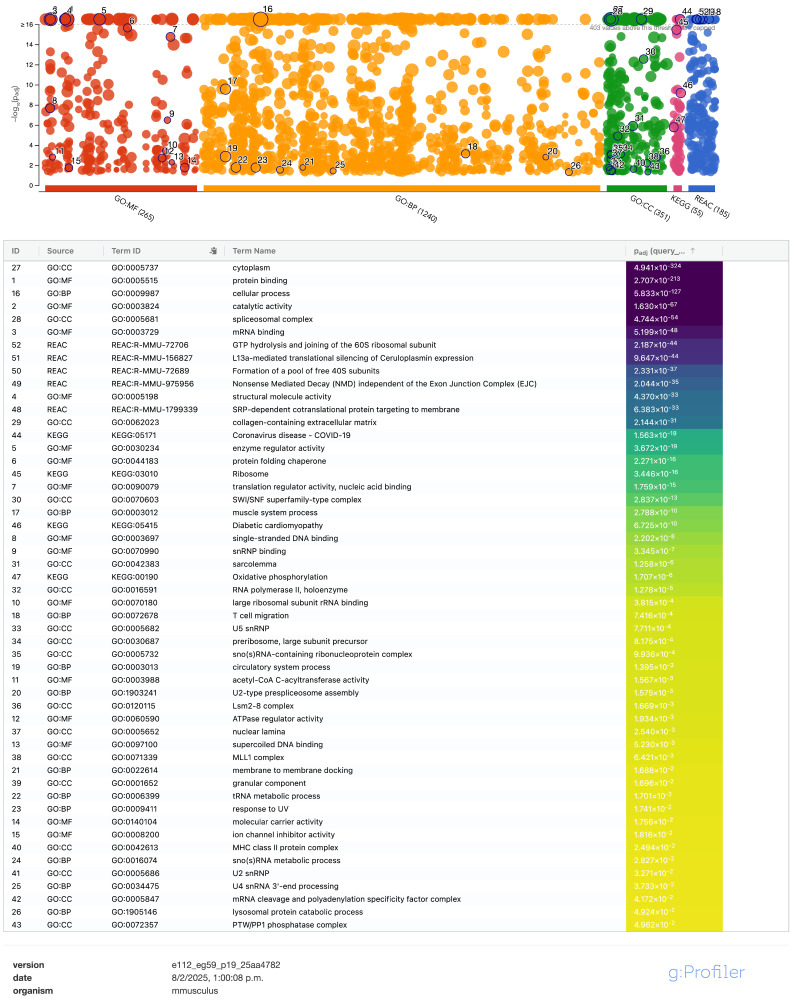
Enrich profiler plot. The number of the 2717 proteins related to each enrichment analysis for each source, Gene Ontology (Biological Process, Cellular Component and Molecular Function), Reactome, and KEGG databases. The *x*-axis shows the number of different enriched processes according to each database, and the *y*-axis plots the significance value obtained in ascending order (−log10 adj-*p*-value). A dashed cutoff at the top of the graph divides the most significant enriched cellular processes, and a black circle with the number of different processes of interest are highlighted according to each database. The list of the representative processes marked is arranged by significance value, labeled by ID (number in the general list of the process), resource (database with which the process was enriched), term ID (identification of the process according to each platform), term name (name of the enriched process according to each database), and adj-*p*-value (significance value of the process), shown with number and color scale and arranged in a descending order.

**Figure 4 pharmaceuticals-18-00242-f004:**
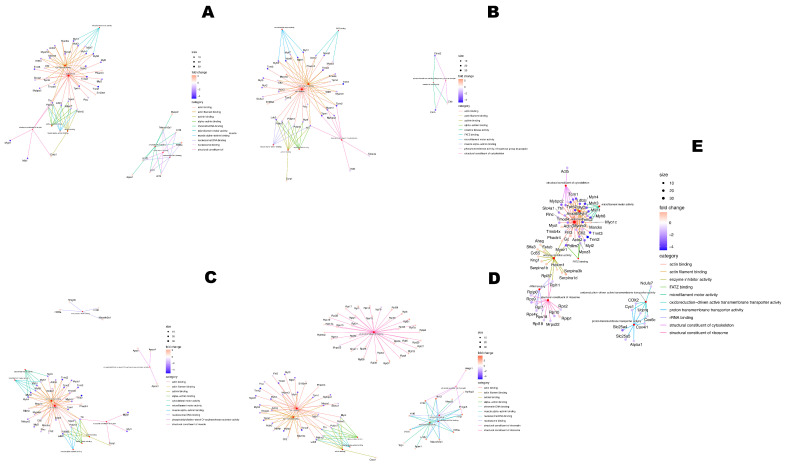
Network comparison via enrichment analysis through Gene Ontology Molecular Function source, from up-regulated and down-regulated proteins from C− versus 5LANM (**A**), 5RINM (**B**), 10LANM (**C**), 10RINM (**D**), and MTX (**E**). Shows term name, molecular function, fold change bar, and node size interactions scale.

**Figure 5 pharmaceuticals-18-00242-f005:**
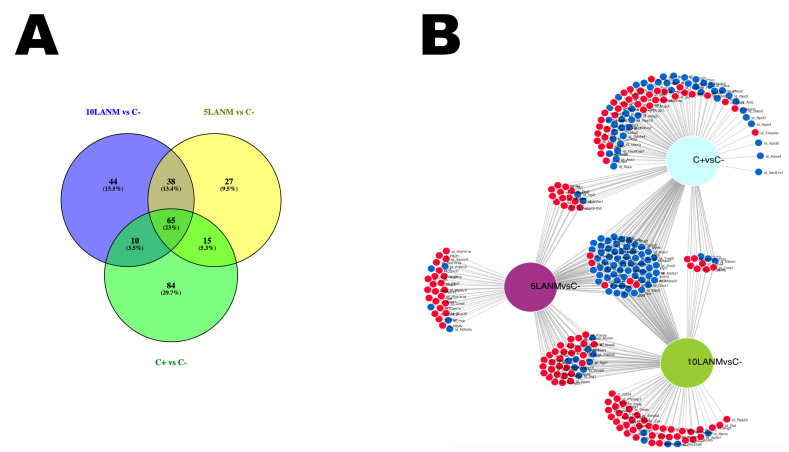
Protein relationship in common between 5LANM, 10LANM, and MTX versus negative control. Venn diagram (**A**) illustrating overlapping circles blue (5LANM), yellow (10LANM), and green (MTX); and network comparison (**B**) indicating up regulation (red dots) or down regulation (blue dots) of proteins in common or unique processes.

**Figure 6 pharmaceuticals-18-00242-f006:**
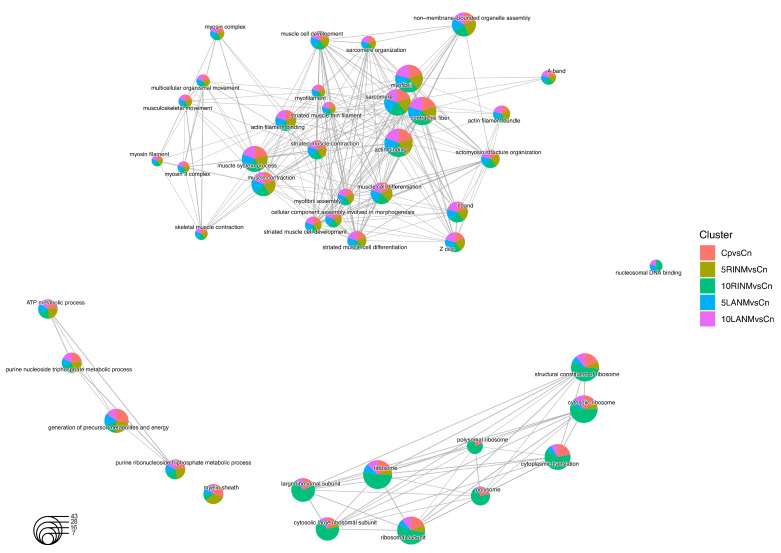
Network comparison via enrichment analysis through Gene Ontology, biological process shared, or specific of dysregulated proteins from C− versus 5LANM, 5RINM, 10LANM, 10RINM, and MTX. Shows cluster color comparison, term name, biological process, and circle scale based on the number of genes.

**Figure 7 pharmaceuticals-18-00242-f007:**
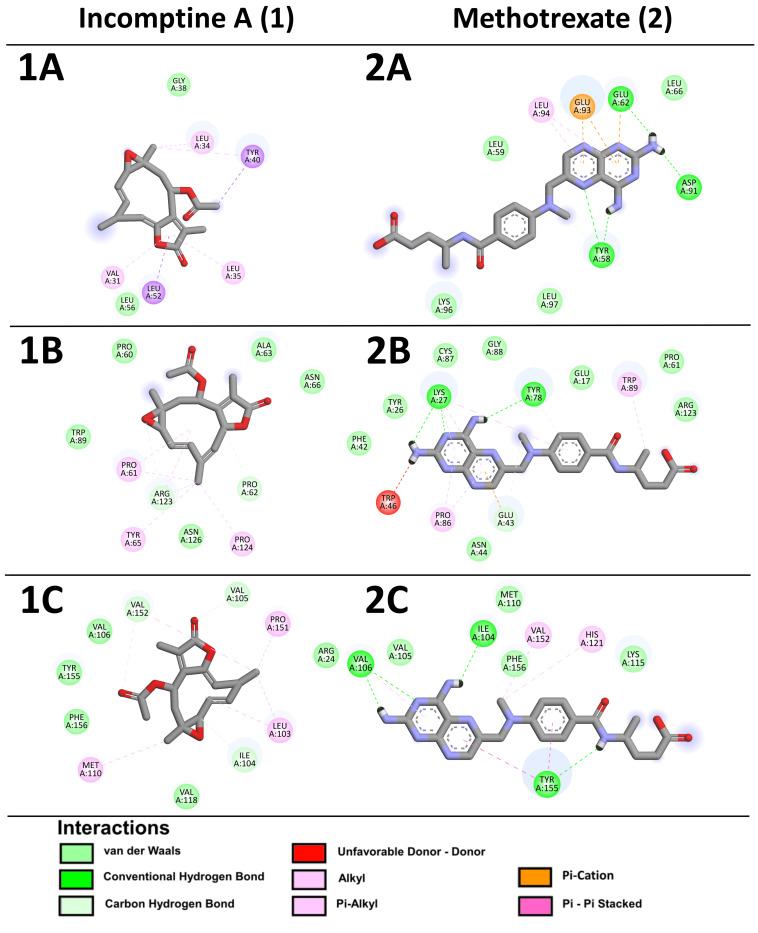
Affinity maps of molecular docking on proteins, 2D representation of interactions between incomptine A (**1**) and methotrexate (**2**) in (**A**) histone H2A type 1-F, (**B**) fragile X mental retardation syndrome-related protein 2, (**C**) DNA-directed RNA polymerase II subunit E, (**D**) 40S ribosomal protein S2, (**E**) signal transducer and activator of transcription 1, (**F**) transforming growth factor beta-1 proprotein, (**G**) TSC22 domain family protein 4, (**H**) apoptosis inhibitor 5, (**I**) cold-inducible RNA-binding protein, (**J**) zinc-alpha-2-glycoprotein, (**K**) alpha-1-acid glycoprotein 1, (**L**) nuclear pore glycoprotein p62, (**M**) beta-enolase, (**N**) caspase-3, and (**O**) caspase-7.

**Table 1 pharmaceuticals-18-00242-t001:** Summary of the differentially expressed from 2717 identified proteins according to their relative quantitation from negative respect treatment and node location.

C− (DMSO) vs.	Down Regulated: FC < 0.67 (<1/1.5)	Up Regulated: FC > 1.5
MTX	111	63
5LANM	76	69
5RINM	**117**	72
10LANM	78	80
10RINM	83	**132**

Data represent the fold change (FC) divided into two categories with the number of differentially expressed proteins. Shows negative control (C−) versus treatments with differentially expressed proteins. No treatment negative control (C−), methotrexate 1.25 mg/kg (MTX), IA 5 mg/kg left axillary node mass (5LANM), IA 5 mg/kg right inguinal node mass (5RINM), IA 10 mg/kg left axillary node mass (10LANM), and IA 10 mg/kg right inguinal node mass (10RINM). Clusters with the highest number of deregulated proteins are highlighted in bold. Analyzed and searched against mouse protein using Maxquant (v2.6.7.0).

**Table 2 pharmaceuticals-18-00242-t002:** ∆G (kcal/mol), Ki (µM), and receptor–ligand interactions of IA and MTX with 15 dysregulated proteins obtained by molecular docking.

Protein	Ligands							
	Incomptine A				Methotrexate (MTX)			
	ΔG	Ki	H-BR	NPI	ΔG	Ki	H-BR	NPI
Histone H2A type 1-F	−4.08	1.03 mM	Gly38, Leu56	Val31, Leu34, Leu35, Tyr40, Leu52	−4.24	778.4 µM	Tyr58, Leu 59, Glu62, Leu66, Asp91, Lys96, Leu97	Glu93, Leu94
Fragile X mental retardation syndrome-related protein 2	−6.83	9.9 µM	Pro60, Pro62, Ala63, Asn66, Trp89, Arg123, Asn126	Pro61, Tyr65, Pro124	−4.99	220.3 µM	Glu17, Tyr26, Lys27, Phe42, Glu43, Asn44, Pro61, Tyr78, Cys87, Gly88, Arg123	Trp46, Pro86, Trp89
DNA-directed RNA polymerase II subunit E	−5.18	160.5 µM	Arg24, Ile104, Val105, Val106, Val118, Val152, Tyr155, Phe156	Leu103, Met110, Pro151	−4.15	904.1 µM	Ile104, Val105, Val106, Met110, Lys115, Tyr155, Phe156	His121, Val152
40S ribosomal protein S2	−5.17	162.6 µM	Ile81, Ser85, Leu86, Pro87, Ile88, Ser161, Ile162, Pro164	Tyr82, Val163	−7.02	1.81 µM	Phe84, Ser85, Leu86, Ile162, Ser245, Lys246, Tyr248, Ser249	Tyr82, Pro87, Ile88, Val163, Pro164
Signal transducer and activator of transcription 1	−4.43	565.7 µM	Val362, Leu363, Phe364, Asn381, Leu383, Gly 384	Lys361, Ile382, His386	−5.12	175.6 µM	Asn233, Val237, Trp239, Lys240, Arg241, Gln243, Gln244, Val318, Val319, Gln322, Ala479, Glu480	Leu453, Pro481
Transforming growth factor beta-1 proprotein	−4.34	653.7 µM	Ala89, Tyr317, His318, Glu362, Pro363, Pro365	Tyr81, Arg85, Val88, Pro314, Leu364, Ile383	−4.72	347.2 µM	Val88, Ala89, Ser92, Tyr299, Gly316, Tyr317, His318, Leu364, Pro365	Pro314, Glu362, Ile383
TSC22 domain family protein 4	−4.37	629.8 µM	Glu101, His103, Ser104	Leu100, Pro102, Phe105	−2.08	29.86 µM	Leu152, Arg153, Pro154, Asn355, Ala356, Ala357, Glu359, Gln360	Pro155
Apoptosis inhibitor 5	−5.19	158.2 µM	Lys404, Val413, Pro445, Val446, Asn496, Tyr497, Glu498	Lys409, Val412, Phe495	−5.1	182.68 µM	Pro358, Ile431, Pro433, Ser434, Tyr435, Lys436	Arg335, His430, Pro432
Cold-inducible RNA-binding protein	−5.92	45.5 µM	Phe49, Asp80, Ala82, Gly83, Lys84, Ser85	Lys7, Phe9, Phe51, Gln81	−4.58	436.7 µM	Asp4, Lys7, Phe9, Gly11, Gly12, Arg47, Phe51, Arg78, Ala82, Gly83, Lys84, Ser85	Phe49
Zinc-alpha-2-glycoprotein	−6.08	35.0 µM	Trp135, Tyr137, Lys167, Trp168, Tyr174, Val175, Arg177	Ile96, Trp154, Ala178	−4.0	436.7 µM	Tyr34, Phe97, Thr100, Leu117, Tyr137, Trp154, Trp168	Arg93, Ile96, Trp135, Tyr174
Alpha-1-acid glycoprotein 1	−5.15	167.8 µM	Val59, Thr65, Glu82, Gln84, Val110, Leu130, Ser143, Tyr145	Tyr45, Phe50, Ile62, Arg108, Phe132	−4.92	246.1 µM	Val27, Pro30, Ile31, Thr32, Thr35, Gln38, Leu119, Ile121, Arg123	Ile20, Ala24, Asn25, Val29, Leu120, Leu122, Met129
Nuclear pore glycoprotein p62	−4.07	1.04 nM	Ile225, Ala226, Thr227, Pro229, Asp394, Ile396, Gln400	Ala228, Leu393, Leu397	−2.35	18.8 mM	Ser238, Leu239, Thr241, Asn459, Lys476, Ile477, Asn479, Ala480, Asp483, Gln486	Cys240, Met482
Beta-enolase	−4.81	298.8 µM	Lys228, Thr229, Gln232, Val240, Asn286, Tyr287	Ile231, Pro237, Pro280	−5.83	53.5 µM	Ser37, Gly38, Glu45, Ala46, Leu47, Glu48, Lys54, Leu58, Gly59, Asn345, Gln346, Arg372	Thr41, Arg50, Lys60
Caspase-3	−5.29	137.7 µM	Glu124, Gly125, Thr140, Ile160, Arg164, Tyr197, Val266	Leu136, Lys137, Tyr195, Met268	−3.38	3.34 mM	Lys137, Ther140, Lys186, Glu190, Tyr195, Tyr197	Cys184, Pro188, Val266, Met268
Caspase-7	−6.12	89.2 µM	Asn148, Lys160, Thr163, Arg187, Glu216, Tyr223, Val292, Met294	Ile159, Phe221	−4.1	995.2 µM	Ala24, Lys25, Pro26, Arg28, Ser29, Phe31, Pro33, Gly188, Thr189	Asp27, Val32, Glu146, Glu147

ΔG: binding energy (kcal/mol^−1^); H-BR: H-binding residues; NPI: nonpolar interactions; Asp: aspartate; Asn: asparagine; Arg: arginine; Gln: glutamine; Lys: lysine; Thr: threonine; Ser: serine; Trp: tryptophan; Leu: leucine; His: histidine; Gly: glycine; Glu: glutamic acid; Ile: isoleucine; Tyr: tyrosine; and Phe: phenylalanine.

**Table 3 pharmaceuticals-18-00242-t003:** Expected physicochemical, pharmacokinetic, and toxicological properties for IA and MTX.

Smiles	**IA**	C=C1C(=O)O[C@@H]2/C=C(/C)CC[C@H]3O[C@@]3(C)C[C@@H](OC(C)=O)[C@H]12				
	**MTX**	CN(Cc1cnc2nc(N)nc(N)c2n1)c1ccc(C(=O)N[C@@H](CCC(=O)O)C(=O)O)cc1				
**Physicochemical Properties**						
		**IA**	**MTX**	**Druglikeness**		
Molecular formula		C_17_H_22_O_5_	C_20_H_22_N_8_O_5_			
Molecular weight		306.35 g/mol	454.44 g/mol		**IA**	**MTX**
TPSA		65.13 Å^2^	210.54 Å^2^	Lipinsky	Yes	Yes
Lipophilicity (LogP)		2.33	−0.32	Ghose	Yes	Yes
Water solubility (LogS)		−2.71	−2.41	Veber	Yes	No, 1 volation
Solubility class		Soluble	Very soluble	Egan	Yes	No, 1 volation
Number of rotating links		2	10	Muegge	Yes	No, 1 volation
Number of H-bond donors		0	5	PAINS	0	0
Number of H-bond acceptors		5	9			
**Pharmacokinetics Properties**						
		**Absorption**			**Metabolism**	
		**IA**	**MTX**		**IA**	**MTX**
Gastrointestinal absorption		High	Low	CYP2C9 substrate	No	No
Hematoencephalic barrier		Yes	No	CYP2D6 substrate	No	No
Caco-2 permeability		High	Low	CYP3A4 substrate	Yes	Yes
*p*-glycoprotein substrate		Yes	Yes	CYP2C9 inhibitor	No	No
*p*-glycoprotein inhibitor		No	No	CYP2D6 Inhibitor	No	No
Log Kp (skin permeation)		−6.89 cm/s	−10.39 cm/s	CYP3A4 Inhibitor	No	No
				CYP1A2 Inhibitor	No	No
				CYP2C19 Inhibitor	No	No
		**Distribution**			**Excretion**	
Mitochondrial		Yes	Yes	CL	4.64 mL/min/Kg	2.41 mL/min/Kg
Protein plasma binding		75.9%	63.4%	T _1/2_	0.78 h	0.39 h
Volume Distribution		1.38 L/Kg	0.32 L/Kg			
**Toxicity**						
		**IA**	**MTX**		**IA**	**MTX**
Hepatotoxicity		Inactive	Active	Carcinogenicity	Inactive	Inactive
Neurotoxicity		Inactive	Active	Immunotoxicity	Inactive	Inactive
Nephrotoxicity		Inactive	Inactive	Mutagenicity	Inactive	Inactive
Respiratory toxicity		Active	Active	Cytotoxicity	Inactive	Inactive
Cardiotoxicity		Inactive	Inactive			
Predicted rats LD_50_		2.68 mol/Kg	3.49 mol/Kg			
Predicted human LD_50_		1330 mg/Kg	3 mg/Kg			
Expected toxicity class *		IV	I			

Predictions were obtained from ADMETlab, SwissADME, admetSAR, and PROTOX web servers. * Toxicity classes are defined according to the Globally Harmonized System of Classification and Labeling of Chemicals (GHS). LD_50_ is expressed in mg/kg. Class I: fatal by ingestion (LD_50_ ≤ 5); Class II: fatal by ingestion (5 < LD_50_ ≤ 50); Class III: fatal by ingestion (50 < LD_50_ ≤ 300); Class IV: fatal by ingestion (300 < LD_50_ ≤ 2000); Class V: fatal by ingestion (2000 < LD_50_ ≤ 5000); and Class VI: fatal by ingestion (LD_50_ > 5000).

## Data Availability

The original contributions presented in this study are included in the article/Appendix A. Further inquiries can be directed to the corresponding author.

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
