# Peer review of "Quantitative Proteomics and Molecular Mechanisms of Non-Hodgkin Lymphoma Mice Treated with Incomptine A, Part II"

_pharmaceuticals, 2025, doi:10.3390/ph18020242_

Round 1
Reviewer 1 Report
Comments and Suggestions for Authors
The authors conducted a comparative proteomic analysis of lymph nodes in mice treated with Incomptine A (IA) or methotrexate, as well as healthy controls, identifying 2,717 proteins, of which 412 were differentially expressed. These proteins were linked to key cellular processes such as chromatin remodeling, metabolism, and necroptosis. The manuscript explores the dose-dependent anticancer effects of IA, emphasizing its potential as a promising secondary metabolite for the treatment of non-Hodgkin lymphoma. The LC-MS/MS methodology described is appropriate for a typical TMT-based quantitative proteomics experiment using nanoflow UPLC and mass spectrometry. Furthermore, the authors demonstrate advanced expertise in bioinformatics tools for proteomic data analysis.
Given that all the data were generated from proteomics analysis, I have the following questions and suggestions regarding the proteomics methodology:
Quantitative Methodology: The authors conducted a TMT-labeled quantitative method in the experiment but referred to it as iTRAQ-based quantitative proteomics in the title and introduction. While TMT (Tandem Mass Tag) and iTRAQ (Isobaric Tags for Relative and Absolute Quantification) are similar isobaric labeling methods, they are not the same in proteomics. Based on the manuscript details, it appears that the authors employed TMT rather than iTRAQ for their proteomic analysis.
Differentially Expressed Proteins: The authors quantified 2,717 proteins using TMT-based quantitative proteomics, identifying 132 upregulated and 117 downregulated proteins. However, the manuscript does not provide statistical P values for these findings. The authors should clarify the criteria used to define proteins as significantly changed, such as fold-change thresholds or additional statistical tests.
Bioinformatic Methodology: In the Bioinformatic Methodology section (line 748), the authors state "Absolute values of protein changes." It would be more accurate to remove the term "absolute," as the TMT method is relative. Additionally, in line 749, the phrase "in base 1.5 of the quotient" is not clear—do the authors mean "proteins with a 1.5-fold change"? Using a logarithmic base of 1.5 is unusual and may get confused.
Raw Data Submission: all raw LC-MS/MS data should be submitted to a recognized protein repository such as PRIDE (Proteomics Identifications Database).
Comments on the Quality of English LanguageSome minor grammatical errors should be addressed to improve the manuscript's presentation.
Author Response
We sincerely thank all reviewers for their thorough reviews and constructive comments on our manuscript. Their valuable suggestions have helped us significantly improve the quality of our article. We have carefully addressed each of their comments as detailed below.
Reviewer 1, all suggestion for the manuscript pharmaceuticals-3329328 were attended and highlihted in red color. Also, the quality of English language of this version was reviewed by all authors and reviewed by a native English speaker to improve the presentation of the manuscript.
Comments and Suggestions for Authors
The authors conducted a comparative proteomic analysis of lymph nodes in mice treated with Incomptine A (IA) or methotrexate, as well as healthy controls, identifying 2,717 proteins, of which 412 were differentially expressed. These proteins were linked to key cellular processes such as chromatin remodeling, metabolism, and necroptosis. The manuscript explores the dose-dependent anticancer effects of IA, emphasizing its potential as a promising secondary metabolite for the treatment of non-Hodgkin lymphoma. The LC-MS/MS methodology described is appropriate for a typical TMT-based quantitative proteomics experiment using nanoflow UPLC and mass spectrometry. Furthermore, the authors demonstrate advanced expertise in bioinformatics tools for proteomic data analysis.
Given that all the data were generated from proteomics analysis, I have the following questions and suggestions regarding the proteomics methodology:
Comments 1: Quantitative Methodology: The authors conducted a TMT-labeled quantitative method in the experiment but referred to it as iTRAQ-based quantitative proteomics in the title and introduction. While TMT (Tandem Mass Tag) and iTRAQ (Isobaric Tags for Relative and Absolute Quantification) are similar isobaric labeling methods, they are not the same in proteomics. Based on the manuscript details, it appears that the authors employed TMT rather than iTRAQ for their proteomic analysis.
Response 1: Thank you for pointing this out. We agree with this comment. Therefore, we made the correct annotation Tandem Mass Tags (TMT) to refer the proteomic analysis, in the revised manuscript this change can be found in page 1, abstract, and line 28; page 1, keywords, and line 42; page 2, paragraph 6, and line 90; and page 27, subtitle, and line 752.
Comments 2: Differentially Expressed Proteins: The authors quantified 2,717 proteins using TMT-based quantitative proteomics, identifying 132 upregulated and 117 downregulated proteins. However, the manuscript does not provide statistical P values for these findings. The authors should clarify the criteria used to define proteins as significantly changed, such as fold-change thresholds or additional statistical tests.
Response2: Thank you for pointing this out. Since Tandem Mass Tags (TMT) is a labeling technique used in quantitative proteomics, by isobaric chemical tags that reflects the abundance of combined samples and allows simultaneous analysis of all samples in one run. This reduces run-to-run variability, increases throughput, and improves the statistical power of the analysis, which result in an accurate quantification. Therefore, there were no need of additional statistical tests, given that the fold-change thresholds were generated from a single value from the raw data, as established in methods, we clustered the lymph nodes as pools of all mice in each group. Furthermore, the complexity and large amount of data generated by TMT analysis were made through sophisticated bioinformatics tools and resources for accurate interpretation and validation of results, data analytical report was performed by the Analytical Service from Creative Proteomics (NY, USA), with p-values smaller than 0.05 in the analysis (where P-value < 0.05 indicates > 95% confidence of a change in protein concentration irrespective of the magnitude of the change) was selected to designate differentially expressed proteins (can be found in page 27, paragraph 3, and line 792). Regarding the fold-change thresholds the criteria text was added (Fold change describes the ratio of two values (expression condition 1)/(expression condition 2). The most common fold-change cutoff at protein level accepted with a biological meaning are defined by 1.5 times up or down [1]; sometimes less rigor fold change cut-off ratio of > 1.32 or < 0.68 [2] or with fold-change >1.2 or <0.83 in abundance [3], can be found in page 28, paragraph 1, and line 803.
Comments 3: Bioinformatic Methodology: In the Bioinformatic Methodology section (line 748), the authors state "Absolute values of protein changes." It would be more accurate to remove the term "absolute," as the TMT method is relative. Additionally, in line 749, the phrase "in base 1.5 of the quotient" is not clear—do the authors mean "proteins with a 1.5-fold change"? Using a logarithmic base of 1.5 is unusual and may get confused.
Response 3: Thank you for pointing this out. We reviewed the writing of the text, for a better understanding. Since the raw data or the fold change cutoff scale from >1.5 to <0.67 in continuous values cannot be used in the analysis process due the software requirements, since ratios are not symmetric around one. Therefore, the log ratio (Log1.5) are the log-of-the-fold-changes i.e. log1.5(condition1/condition2). log ratios are used/plotted in graphs for the exploratory analysis of pathways and functional enrichment of cellular processes in each of the different treatments, generating a log ratio values matrix with each protein, as those are nicer to show because they center around 0, giving reductions a negative value and increments a positive value, log ratios are symmetric around zero [4,5]. The log ratio data matrix was used to feed the algorithms for the analysis of cellular process enrichment networks and signaling pathways. Can be found in page 28, paragraph 2, and line 812).
Comments 4: Raw Data Submission: all raw LC-MS/MS data should be submitted to a recognized protein repository such as PRIDE (Proteomics Identifications Database).
Response 4: Thank you for pointing this out. Since the TMT analysis were performed by the Analytical Service from Creative Proteomics (NY, USA) as a service, Raw files contain raw mass spectrometry spectra at any format (raw, mzML, psm, and mzid), were not requested. Analytical reports were obtained in *.xlms format, which was enough to continue with the corresponding analyzes. Therefore, at present moment it will not be possible to carry out the upload of raw LC-MS/MS data at the protein repository. However, we will make the corresponding request to obtain the files, to put the files in said repository. We hope that this is not a reason for not continuing with the publication of the results. Can be found in page 27, paragraph 4, and line 792).
Comments 5: Comments on the Quality of English Language
Some minor grammatical errors should be addressed to improve the manuscript's presentation.
Response 5: Thank you for pointing this out. The quality of English language of this version was reviewed by all authors and reviewed by a native English speaker to improve the presentation of the manuscript.
Thank for your consideration. We sincerely hope that the changes made will clarify the doubts and comments; and meet the expectations of the reviewers.
- Gautam, P.; Nair, S.C.; Gupta, M.K.; Sharma, R.; Polisetty, R.V.; Uppin, M.S.; Sundaram, C.; Puligopu, A.K.; Ankathi, P.; Purohit, A.K.; et al. Proteins with altered levels in plasma from glioblastoma patients as revealed by iTRAQ-based quantitative proteomic analysis. PLoS One 2012, 7, e46153, doi:10.1371/journal.pone.0046153.
- Zhang, Y.; Wang, Y.; Li, S.; Zhang, X.; Li, W.; Luo, S.; Sun, Z.; Nie, R. ITRAQ-based quantitative proteomic analysis of processed Euphorbia lathyris L. for reducing the intestinal toxicity. Proteome Sci 2018, 16, 8, doi:10.1186/s12953-018-0136-6.
- Li, S.; Su, X.; Jin, Q.; Li, G.; Sun, Y.; Abdullah, M.; Cai, Y.; Lin, Y. iTRAQ-Based Identification of Proteins Related to Lignin Synthesis in the Pear Pollinated with Pollen from Different Varieties. Molecules 2018, 23, doi:10.3390/molecules23030548.
- Sivanich, M.K.; Gu, T.J.; Tabang, D.N.; Li, L. Recent advances in isobaric labeling and applications in quantitative proteomics. Proteomics 2022, 22, e2100256, doi:10.1002/pmic.202100256.
- Bailey, A. Chapter 5 Transforming and visualising proteomics data. Available online: https://ab604.github.io/docs/bspr_workshop_2018/transform.html (accessed on nov 2024).
Reviewer 2 Report
Comments and Suggestions for Authors
The article titled "Understanding the molecular mechanisms of Balb/c non-Hodgkin lymphoma mice treated with Incomptine A through iTRAQ-based quantitative proteomics: bioinformatics approaches, part II." provides evidence that Incomptine A (IA) is a secondary metabolite with potential as an anticancer agent for the treatment of non-Hodgkin lymphoma. The authors performed a series of experiments, used isobaric tags for relative and absolute quantitation (iTRAQ) based on LC-MS/MS method and a molecular docking experiment to figure out the protein changes of two IA concentrations (5 & 10 mg/kg) in an in vivo male Balb/c mice model inoculated with U-937 cells, compared to positive control methotrexate. The results are interesting and potentially could contribute to the research field. However, there are some significant issues with this article that need to be addressed:
1. The title should be shortened and concise.
2. In the abstract, more methodological information should be incorporated.
3. Keywords should be within 5-6. In the present MS, there are 10 keywords.
4. The introduction section of the manuscript requires more background information. The authors should discuss the antitumor activities and molecular mechanism of Incomptine A. Key proteins associated with non-Hodgkin lymphoma (NHL) should be discussed (i.e., BCL-2, Cyclin D1, ALK, BCL-6). Need to highlight the research gap and significance of the study appropriately. Please carefully check the last paragraph of the introduction and remove the repeated texts (lines 91-96) that were already mentioned in the abstract. Please clearly mention the objectives of the study.
5. Figures should be more visible and comprehensive. Please enlarge the texts (i.e., legends, axis titles) on the figures (figures 2, 3, 4, 5, 6).
6. Results and discussion section should be more concise and focused. Please compare your results with previous reports. In lines, 617-619, the authors claimed that their “findings may provide information on benefit on patient's diagnosis, treatment, or prognosis, which is crucial for those who suffer from the disease”. How will their findings help to diagnosis of the non-Hodgkin lymphoma (NHL) disease? Please discuss briefly.
7. Try to add appropriate references for the methods (i.e., cell culture conditions; anti-lymphoma treatments; sample preparation for iTRAQ-based proteomic analysis; Nano LC-MS/MS Analyses; In Silico Studies, molecular docking). Why did the concentrations (doses) of IA choose 5 & 10 mg/kg in this study? Is there any justification? Please discuss the detail QA and QC for HPLC and LC-MS/MS Analyses.
8. The conclusion should link with the aim/objectives of the study.
9. It is necessary for this manuscript to address the limitations of the study.

Author Response
We sincerely thank all reviewers for their thorough reviews and constructive comments on our manuscript. Their valuable suggestions have helped us significantly improve the quality of our article. We have carefully addressed each of their comments as detailed below.
Reviewer 2, all suggestion for the manuscript pharmaceuticals-3329328 were attended and highlihted in red color.
Comments and Suggestions for Authors
The article titled "Understanding the molecular mechanisms of Balb/c non-Hodgkin lymphoma mice treated with Incomptine A through iTRAQ-based quantitative proteomics: bioinformatics approaches, part II." provides evidence that Incomptine A (IA) is a secondary metabolite with potential as an anticancer agent for the treatment of non-Hodgkin lymphoma. The authors performed a series of experiments, used isobaric tags for relative and absolute quantitation (iTRAQ) based on LC-MS/MS method and a molecular docking experiment to figure out the protein changes of two IA concentrations (5 & 10 mg/kg) in an in vivo male Balb/c mice model inoculated with U-937 cells, compared to positive control methotrexate. The results are interesting and potentially could contribute to the research field. However, there are some significant issues with this article that need to be addressed:
Comments 1: The title should be shortened and concise.
Response 1: Accordingly, to your suggestion we have shortened and update the title to: “Quantitative proteomics and molecular mechanisms of non-Hodgkin lymphoma mice treated with Incomptine A, part II”, this change can be found in page number 1, title, and line 2).
Comments 2: In the abstract, more methodological information should be incorporated.
Response 2: Agree. Thank you for pointing this out. To clearly understand the core content of our publication, we modify to a structured abstract and add more methodological information. This change can be found in page number 1, Abstract, and line 26).
Comments 3: Keywords should be within 5-6. In the present MS, there are 10 keywords.
Response 3: Thank you for pointing this out. We have revised the pertinent keywords to be more specific to the article and the subject discipline (Keywords: incomptine A; non-Hodgkin lymphoma; proteome TMT-based; molecular docking, necroptosis). This change can be found in page number 1, keywords, and line 42). We want to mention that in the Instructions for Authors of the pharmaceutical journal, it is noted that they allow three to ten pertinent keywords. We sincerely appreciate your thorough review and constructive comments on our manuscript.
Comments 4: The introduction section of the manuscript requires more background information. The authors should discuss the antitumor activities and molecular mechanism of Incomptine A. Key proteins associated with non-Hodgkin lymphoma (NHL) should be discussed (i.e., BCL-2, Cyclin D1, ALK, BCL-6). Need to highlight the research gap and significance of the study appropriately. Please carefully check the last paragraph of the introduction and remove the repeated texts (lines 91-96) that were already mentioned in the abstract. Please clearly mention the objectives of the study.
Response 4: Thank you for pointing this out. We remove the repeated texts in the last paragraph of the introduction. We sincerely appreciate your thorough review and constructive comments on our manuscript. We have revised the background information in the revised manuscript to make it more concise and highlight the important points and objectives. Discuss briefly Key proteins BCL-2, BCL-6 and MYC, can be found in page number 23, paragraph 1, and line 545.
Comments 5: Figures should be more visible and comprehensive. Please enlarge the texts (i.e., legends, axis titles) on the figures (figures 2, 3, 4, 5, 6).
Response 5: Thank you for pointing this out. We have improved to the extent that the programs allow us to emphasize this point. Due to the amount of data, when changing the size names, the networks become saturated and indistinguishable. This can be found in page 4, figure 2, and line 148; in page 5, figure 3, and line 173; in page 7, figure 4, and line 219; in page 9, figure 5, and line 313; and in page 11, figure 6, and line 396.
Comments 6: Results and discussion section should be more concise and focused. Please compare your results with previous reports. In lines, 617-619, the authors claimed that their “findings may provide information on benefit on patient's diagnosis, treatment, or prognosis, which is crucial for those who suffer from the disease”. How will their findings help to diagnosis of the non-Hodgkin lymphoma (NHL) disease? Please discuss briefly.
Response 6: Thank you for pointing this out. We agree with this comment. Therefore, in the results and discussion section we added and changed paragraphs to be more clearly, concise and focused. These changes can be found in page number 3, paragraph 1, and line 110; in page number 19, paragraph 2, and line 431; in page number 23, paragraph 1, and line 545; in page number 24, paragraph 1, and line 602; in page number 24, paragraph 3, and line 619; in page number 24, paragraph 4, and line 640. We paraphrase and reword the mentioned phrase so that what we wrote is understood. This change can be found in page number 25, paragraph 3, and line 676.
Comments 7: Try to add appropriate references for the methods (i.e., cell culture conditions; anti-lymphoma treatments; sample preparation for iTRAQ-based proteomic analysis; Nano LC-MS/MS Analyses; In Silico Studies, molecular docking). Why did the concentrations (doses) of IA choose 5 & 10 mg/kg in this study? Is there any justification? Please discuss the detail QA and QC for HPLC and LC-MS/MS Analyses.
Response 7: Thank you for pointing this out. New references were added to the section to reinforce methodology, can be found in page 25, paragraph 4, and line 700; in page 26, paragraph 1, and line 707; in page 26, paragraph 3, and line 728; in page 27, paragraph 1, and line 751; in page 27, paragraph 2, and line 770; in page 27, paragraph 3, and line 782; in page 27, paragraph 4, and line 792; in page 28, paragraph 1, and line 803; in page 28, paragraph 1, and line 793; in page 28, paragraph 2, and line 819; in page 28, paragraph 2, and line 828; in page 28, paragraph 3, and line 848; in page 29, paragraph 1, and line 857; in page 29, paragraph 2, and line 863-864.
The chosen concentrations are supported by a previous in vitro work, that has been developed with the IA. Can be found in page 2, paragraph 5, and line 83; in page 3, paragraph 1, and line 110; and in page 26, paragraph 3, and line 728.
Quality Assurance (QA) and Quality Control (QC) were carried out and guaranteed by Creative Proteomics Analytical Service (NY, USA). We know how important they are to ensure the reliability and accuracy of HPLC and LC-MS/MS analyses and are also guaranteed by the chemistry of the TMT6plex Isobaric Label Reagent Set, Pierce Quantitative Colorimetric Peptide Assay (Thermo Fisher Science). Since this is a standardized service, we do not consider it necessary to write about it in detail or discuss it. We sincerely appreciate your thorough review and constructive comments on our manuscript.
Comments 8: The conclusion should link with the aim/objectives of the study.
Response 8: Thank you for pointing this out. Accordingly, we modified the conclusions to emphasize this point. This change can be found in page number 29, Conclusions, and line 891).
Results from differentially expressed protein levels examined in lymph nodes of an in vivo mouse model confirm the previously observed dose-dependent effect of IA as a secondary metabolite with important potential as an anticancer agent for the treatment of NHL, showing that drug type or anatomical location influences treatment response by triggering modifications of pathways or processes, exclusive or shared between treatments, such as chromatin remodeling, transcription, translation, energy metabolism, oxidative phosphorylation, glycolysis/gluconeogenesis, cell proliferation, cytoskeletal organization, cell death compromised by necroptosis and apoptosis, which define cell fate.
Although methotrexate is an important drug used in cancer treatment, IA is a good alternative which, based on in silico analysis, is likely to have better activity on different cancer-related targets, has better solubility, lower toxicity effects, a higher LD50 which translates into a wider range of doses to be used and, in addition to its pharmacological effect, is a good candidate for the development of a new anticancer drug.
Further research is needed to increase understanding of the role of IA as an antitumor drug, to improve outcomes and survival of NHL patients, and to understand how it will affect pathways leading to cell death proteins. IA promises to be a potentially safer and more effective treatment to improve outcomes, reduce toxicities and enhance survival in patients with NHL, initially targeting histones and transcription factors that will affect cell death proteins.
Comments 9: It is necessary for this manuscript to address the limitations of the study.
Response 9: Agree. Thank you for pointing this out. We have, accordingly, add the section strengths and limitations to emphasize this point. This change can be found in page 30, paragraph 2, and line 912.
Strengths and limitations: Tandem Mass Tags (TMT) is a labeling technique used in quantitative proteomics, applicable to global proteomics and targeted analysis of post-translational modifications, involving the use of isobaric chemical tags (designed to be resistant to fragmentation) that consist of a mass reporter region, a mass normalization region, and a peptide reactive group. The intensity of these reporter ions reflects the relative abundance of the respective peptides in the samples, combined and analyzed together. The main advantages of the TMT method are its expanded multiplexing capability, allowing for simultaneous analysis of different samples in one run. This reduces run-to-run variability, increases throughput, and improves the statistical power of the analysis, which can result in a more accurate quantification. TMT also has its challenges. As ratio compression due to co-isolation and co-fragmentation of peptides can be an issue, potentially leading to inaccurate quantification. TMT is also more expensive than other quantitative proteomics techniques, like label-free quantitation, which may limit its application in some cases. Furthermore, the complexity and large amount of data generated by TMT analysis require sophisticated bioinformatics tools and resources for accurate interpretation and validation of results. Another challenge around in silico prediction is the application domain or validation techniques of the models. These lead us to new research such as computational systems toxicology or data integration, which could be used in drug discovery such as AI, evaluating risks and usefulness so that doctors have compounds against NHL at their disposal.
Thank for your consideration. We sincerely hope that the changes made will clarify the doubts and comments; and meet the expectations of the reviewers.
Reviewer 3 Report
Comments and Suggestions for Authors
Dear authors, the manuscript shows some strength points such as:
1) The in vivo testing.
2) the comprehensive proteome studies.
However, some major and minor corrections should be considered.
Major Corrections:
a) a dynamic simulation study should be added at least for 50 ns, as docking simulation only provides a snapshot of the molecular binding (this reference could help). 10.1021/acs.jmedchem.2c01818
b) An ADMET study should be added to evaluate the drug likeness of the selected candidate and compare it to methotrexate.
c) a PAINS analysis should be done to detect any false positives ( this reference could help /10.1016/j.bioorg.2021.105054).
Minor corrections:
a) a section should be added to explain why methotrexate was selected as a positive control.
b) the resolution of all figures should be enhanced.
Author Response
We sincerely thank all reviewers for their thorough reviews and constructive comments on our manuscript. Their valuable suggestions have helped us significantly improve the quality of our article. We have carefully addressed each of their comments as detailed below.
Reviewer 3, all suggestion for the manuscript pharmaceuticals-3329328 were attended and highlihted in red color.
Comments and Suggestions for Authors
Dear authors, the manuscript shows some strength points such as:
1) The in vivo testing.
2) the comprehensive proteome studies.
However, some major and minor corrections should be considered.
Major Corrections:
Comments 1: a) a dynamic simulation study should be added at least for 50 ns, as docking simulation only provides a snapshot of the molecular binding (this reference could help). 10.1021/acs.jmedchem.2c01818
Response 1: Thank you for pointing this out and your thoughtful suggestion, we are working on it, dynamic molecular studies are being carried out to obtain more information as a complement to this research, however that information is going to be published in another manuscript that we are already writing. We sincerely appreciate your thorough review and constructive comments on our manuscript.
Comments 2: b) An ADMET study should be added to evaluate the drug likeness of the selected candidate and compare it to methotrexate.
Response 2: Agreed. We have therefore performed the ADMET study and added it to the manuscript, this change can be found in page number 19, results paragraph 2, and line 431; in page number 24, discussion paragraph 3, and line 619; in page number 29, methods paragraph 2, and line 858; and in page number 30, conclusion paragraph 3, and line 905.
Comments 3: c) a PAINS analysis should be done to detect any false positives (this reference could help /10.1016/j.bioorg.2021.105054).
Response 3: Agreed. Pan-assay interfering compounds (PAINS) are useful for medicinal chemists to optimize compounds, that may benefit our health, such as IA. We have therefore added PAINS analysis to the manuscript. This change can be found in page number 19, results paragraph 2, and line 431; in page number 24, discussion paragraph 3, and line 619; in page number 29, methods paragraph 2, and line 858; and in page number 30, conclusion paragraph 3, and line 905.
Minor corrections:
Comments 4: a) a section should be added to explain why methotrexate was selected as a positive control.
Response 4: Thank you for pointing this out. We agree with this comment. Therefore, we add a paragraph. NHL represents a broad spectrum of diseases that are one of the main causes of morbidity and mortality worldwide. Chemotherapy is an effective treatment against NHL, used alone or in combination with methotrexate, cyclophosphamide and doxorubicin. However, to date, all the drugs used induce various side effects, as is the case with MTX, which is the drug of choice for the treatment of NHL in Mexico [1-3]. This change can be found in discussion, page number 24, paragraph 1, and line 602.
Comments 5: b) the resolution of all figures should be enhanced.
Response 5: Thank you for pointing this out. We have improved to the extent that the programs allow us to emphasize this point. Due to the amount of data, when changing the size names, the networks become saturated and indistinguishable. This can be found in page 4, figure 2, and line 148; in page 5, figure 3, and line 173; in page 7, figure 4, and line 219; in page 9, figure 5, and line 313; and in page 11, figure 6, and line 396.
Thank for your consideration. We sincerely hope that the changes made will clarify the doubts and comments; and meet the expectations of the reviewers.
- Calzada, F.; Ramirez-Santos, J.; Valdes, M.; Garcia-Hernandez, N.; Pina-Jiménez, E.; Ordoñez-Razo, R.M. Evaluation of Acute Oral Toxicity, Brine Shrimp Lethality, and Antilymphoma Activity of Geranylgeraniol and Annona macroprophyllata Leaf Extracts. Revista Brasileira de Farmacognosia 2020, 30, 301-304, doi:10.1007/s43450-020-00014-8.
- Howard, S.C.; McCormick, J.; Pui, C.H.; Buddington, R.K.; Harvey, R.D. Preventing and Managing Toxicities of High-Dose Methotrexate. Oncologist 2016, 21, 1471-1482, doi:10.1634/theoncologist.2015-0164.
- Velazquez-Dominguez, J.; Marchat, L.A.; Lopez-Camarillo, C.; Mendoza-Hernandez, G.; Sanchez-Espindola, E.; Calzada, F.; Ortega-Hernandez, A.; Sanchez-Monroy, V.; Ramirez-Moreno, E. Effect of the sesquiterpene lactone incomptine A in the energy metabolism of Entamoeba histolytica. Exp Parasitol 2013, 135, 503-510, doi:10.1016/j.exppara.2013.08.015.
Round 2
Reviewer 1 Report
Comments and Suggestions for Authors
Submitting raw data to a recognized repository is now widely required for proteomics publications. Most journals in proteomics and related fields mandate data submission. Authors can request the Creative Proteomics service to upload the files to PRIDE, MassIVE, or other repositories, or they can upload the data themselves. This process is typically very quick, usually taking just a couple of hours.
Author Response
Reviewer 1, Thank you very much for taking the time to review this manuscript. We sincerely appreciate your thorough review and constructive comments on our manuscript. Your valuable suggestions have helped us to significantly improve the quality of our article. We have carefully addressed your comment as detailed below for the manuscript pharmaceuticals-3329328 which was attended to and highlighted in red color.
Comments and Suggestions for Authors
Comments 1: Submitting raw data to a recognized repository is now widely required for proteomics publications. Most journals in proteomics and related fields mandate data submission. Authors can request the Creative Proteomics service to upload the files to PRIDE, MassIVE, or other repositories, or they can upload the data themselves. This process is typically very quick, usually taking just a couple of hours.
Response 1: Thank you for pointing this out. We agree with this comment. Therefore, since the last review you made, we are aware of the need to upload the raw files to the PRIDE platform, then we made the corresponding request to the company Creative Proteomics, through the intermediary company in our country that provided us with the service.
We are currently waiting for the response, mainly if they still have the files, since the study was carried out 2 years ago. In any case, we have at our disposal the file that they reported to us in Excel with all the peptides and values, which we can be provide to anyone who requests it, via email.
I would like to tell you that due to the journal's calendar for the moment we cannot stop the process to upload the files to the repository (because for now we do not have them in our hands), but we promise to do so once we have them available.
I would also like to take this opportunity to tell you that we have four other manuscripts derived from this study, and for this reason we would like to reserve a little that the data to be public, while we publish.
Accordingly, we made the annotation in the revised manuscript: “The raw material will be uploaded to the PRIDE repository once pending analyses are completed and published, or if available, the DOI link will be added to the drive folder (previously annotated) where it can be consulted, or the DOI or the complete Excel file report with all peptides and values from Creative Proteomics can be requested from the corresponding authors”, this change can be found on page 30, Supplementary Materials paragraph 3, line 933.
Thank for your consideration. We sincerely hope that the changes made will clarify the doubts and comments; and meet the expectations of the reviewers.
Reviewer 2 Report
Comments and Suggestions for Authors
The revised manuscript is significantly improved. It can be accepted after the figures' (2, 4, 6) visibility issues are solved.
Author Response
Reviewer 2, Thank you very much for taking the time to review this manuscript. Your valuable suggestions have helped us significantly improve the quality of our article. We have carefully addressed your comment as detailed below for the manuscript pharmaceuticals-3329328 was attended and highlighted in red color.
Comments and Suggestions for Authors
Comments 1: The revised manuscript is significantly improved. It can be accepted after the figures' (2, 4, 6) visibility issues are solved.
Response 1: Thank you for pointing this out. We have noticed that this is an issue during the transfer of information to the MDPI Susy platform, so when generating the file in PDF format, some layer of information is lost. For this reason, we improved the quality and to the best resolution of the figures (2, 4 y 6) by exporting them in TIFF format. In any case, we will coordinate with the publisher at the editorial department, so that in the final version they remain with the pertinent visibility and comprehensive.
The reviewer can consult the files directly in the Figures, Graphs, Images section. We increased the size as much as the programs allowed us to emphasize this point, without saturating the names and networks, due to the amount of data. The figures were replaced on page 4, figure 2 and line 148; on page 7, figure 4 and line 219; and on page 11, figure 6 and line 396. This increases the size of the word file to more than 100 mb.
We are also adding the corresponding images to this section, since we could not add any attached file. If the figures are not shown in this box, please refer to the attached files.
Figure 2. Network visualization from the 2,717 proteins in a computational representation through Gene Ontology with enrichment analysis of Molecular Function subontology. It shows identified categories, and node size in range 40-160 interactions.
Figure 4. Network comparison via enrichment analysis through Gene Ontology Molecular Function source, from up regulated and down regulated proteins from C- versus 5LANM (4A), 5RINM (4B), 10LANM (4C), 10RINM (4D), and MTX (4E). Shows term name, molecular function, fold change bar and node size interactions scale.
Figure 6. Network comparison via enrichment analysis through Gene Ontology, biological process shared or specific of dysregulated proteins from C- versus 5LANM, 5RINM, 10LANM, 10RINM, and MTX. Shows cluster color comparison, term name, biological process, and circle scale based on the number of genes
Thank for your consideration. We sincerely hope that the changes made will clarify the doubts and comments; and meet the expectations of the reviewer.
Sincerely yours,
Dr. Prof. Normand García Hernández, Corresponding author.

Round 3
Reviewer 1 Report
Comments and Suggestions for Authors
The repositories allow the data to be uploaded but remain private until publication. Authors can share access with reviewers during the peer review process using temporary credentials. Authors are required to submit raw data to a public repository before their proteomics manuscript is accepted for publication. This requirement aligns with the standards of the proteomics research community and scientific journals.
Author Response
Dear Reviewer:
We sincerely thank you for your thorough reviews and constructive comments on our manuscript pharmaceuticals-3329328. Your valuable suggestions have helped us significantly improve the quality of our article. We have carefully addressed your comment as detailed below and highlighted in red color.
Comments and Suggestions for Authors
Comments 1: The repositories allow the data to be uploaded but remain private until publication. Authors can share access with reviewers during the peer review process using temporary credentials. Authors are required to submit raw data to a public repository before their proteomics manuscript is accepted for publication. This requirement aligns with the standards of the proteomics research community and scientific journals.
Response 1: Thank you for bringing this to our attention. We appreciate your concern; however, we disagree due to the following aspects:
1.- Since the first time we were asked to upload the raw files to a repository, we contacted the service provider in our country (Rilab Recursos Integrales Para Laboratorio S.A de C.V), which was the link with Creative Proteomics, and they have just responded to our request. We attach it. And according to their policies, they delete the files one year after processing the samples. Our assays are 4 years old, and it seems that they are going to give us the files of another in vitro experiment (already published) that they managed to recover, and from which we will make the corresponding upload to PRIDE.
2.- We understand that sending raw data to a recognized repository is convenient, that the process is usually very quick and normally only takes a couple of hours, and while this may be a requirement for other journals, we do not currently have the raw files, we do not believe and hope that this is not a reason to stop publication. Considering that the Aims and Scope of Pharmaceuticals (ISSN 1424-8247) is an international scientific journal of medicinal chemistry and related drug sciences, with comprehensive theoretical and experimental details, and our manuscript is intended to aspects involved in drug discovery and development (structure-property correlations, molecular modeling, animal experimentation, drug targeting, dosage, and bioinformatics).
3.- As is well known, 15-20 years ago the need to store raw data was controversial and data reproducibility was a concern, initiated by HUPO, so people started to request that raw files be uploaded to a public repository with unrestricted access. This was useful for re-analyzing the raw data, for data mining or multi‐omics integration. Nowadays, in the proteomics community, this is no longer considered a problem, because the quality of the generated data has improved incredibly. In the science of studying large numbers of proteins, proteomic studies are performed in laboratories all over the world, the technology of mass spectrometry (MS) has developed over many years, such that it is now possible to identify and quantify many hundreds, or even thousands, of proteins simultaneously in one type of sample compared with another. Data sets produced by MS, running to many Gigabytes for a single sample analyzed, the raw files are processed, often in two stages by different software packages that first identify and then quantify the proteins that were analyzed by the instrument. Nowadays raw data files are no longer encoded in a vendor-specific data format, so new software can be used to analyze the data in an optimal way, allowing from the generated files their re-use for integration, data visualization and interpretation of data from other studies, improving our knowledge about genomes and biological systems, and improving software tools in this field.
4.- Core facilities and research infrastructures in the field of proteomics have become a fundamental element in establishing high-quality service standards, enabling access to cutting-edge specialized proteomics technologies supported by a broad computational and multiomics spectrum in basic research. With appropriate analytical protocols to obtain data that comply with best practices, the implementation of quality assessment measures and quality controls commonly accepted in the generation of research data in proteomics, with an analytical workflow that offers the robustness of the applied analytical approach without compromising analytical rigor. The proteomic data report (Analytical Service Report and database CPJS01082001, that was added in the link of the drive), which we have received and are providing, adheres to the journal guidelines and contains data quality metrics that serve for any intra- and inter-laboratory and guarantee the technical quality, reproducibility, comparability and integrity of the data. Our publication provides sufficient detail in terms of methodology and data presented to allow readers and reviewers to assess the validity and reproducibility of the data as well as the significance of their observations. This information includes a description and justification of the overall experimental design (technical and biological replicates, sample size, etc.). In addition, we clearly indicate and justify detailed information on how the raw mass spectrometry data were converted to a database searchable format, the search engine used in data processing, the databases, the scoring functions, how they were calculated, and what methods were used to infer significance, generate similar data, and arrive at the same conclusion, even with an identical sample for analysis. We present the description of the methodology and data analysis in sufficient detail to comply with good publication practices.
5.- Considering that two of the three peer reviewers who have evaluated our manuscript are already satisfied with the current version, and since your comment is considered by you as a minor change, we strongly request that our manuscript be considered for further publication. Thank you very much for your understanding.
Sincerely,
Dr. Prof. Normand García Hernández, Corresponding author.
